# Coupling numerical models of deltaic wetlands with AirSWOT, UAVSAR, and AVIRIS-NG remote sensing data

Luca Cortese[1], Carmine Donatelli[1,2], Xiaohe Zhang[1], Justin A. Nghiem[4], Marc Simard[3], Cathleen E. Jones[3], Michael Denbina[3], Cédric G. Fichot[1], Joshua P. Harringmeyer[1], and Sergio Fagherazzi[1]

[1]Department of Earth and Environment, Boston University, Boston, MA, USA
[2]Department of Civil, Architectural and Environmental Engineering, University of Texas at Austin, TX, USA
[3]Jet Propulsion Laboratory, California Institute of Technology, Pasadena, CA, USA.
[4]Division of Geological and Planetary Sciences, California Institute of Technology, Pasadena, CA, USA.

**Correspondence:** Luca Cortese (lucacort@bu.edu), Xiaohe Zhang (zhangbu@bu.edu)

**Abstract.** Coastal marsh survival relies on the ability to increase their elevation and offset sea level rise. It is therefore important to realistically model sediment fluxes between marshes, tidal channels, and bays since sediment availability controls accretion. Traditionally, numerical models have been calibrated and validated using in-situ measurements at few locations within the domain of interest. These datasets typically provide temporal information but lack spatial variability. This paper explores the potential of coupling numerical models with high resolution remote sensing imagery. Products from three sensors from the NASA Delta-X airborne mission are used. UAVSAR provides vertical water level change on the marshland and was used to adjust the bathymetry and calibrate water fluxes over the marsh. AirSWOT yields water surface elevation within bays, lakes and channels and was used to calibrate the Chezy bottom friction coefficient. Finally, imagery from AVIRIS-NG provide maps of total suspended solids (TSS) concentration that were used to calibrate sediment parameters of settling velocity and critical shear stress for erosion. Three numerical models were developed at different locations along coastal Louisiana using Delft3D. The coupling enabled a spatial evaluation of model performance not possible using simple point measurements. Overall, the study shows that calibration of numerical models and their general performance will greatly benefit from remote sensing.

## 1 Introduction

Coastal marshes are among the most important and functional ecosystems on Earth, as they are able to buffer and protect from storm surges and winds (Farber, 1987; Möller et al., 2014; Haddad et al., 2016; Peter Sheng et al., 2022; Temmerman et al., 2023), store carbon (Saintilan et al., 2013; Nahlik and Fennessy, 2016; Rogers et al., 2019), and offer natural habitat to wildlife (Galbraith et al., 2002; Minello et al., 2003). However, it has been estimated that since the 1900s around 50% of coastal wetlands have been lost (Nicholls, 2004). The resilience of present coastal wetlands is altered by accelerated sea level rise (Cahoon et al., 2006; Spencer et al., 2016; Schuerch et al., 2018), enhanced subsidence due to groundwater and oil

extraction (Syvitski et al., 2009), and depleted sediment supply to the coast as a result of extensive river damming (Syvitski et al., 2005). Physically based models are necessary to understand and predict the response of coastal wetlands to such external drivers (Fagherazzi et al., 2020).

Incoming and outgoing sediment fluxes are key for the evolution of coastal marshes in a climate change scenario (Ganju et al., 2017). Consequently, it is imperative that numerical models accurately solve the hydrodynamic and sediment transport processes in the coastal area. Currently, scientists use stations that measure local parameters such as water levels, temperature, salinity, and sediment concentration to validate physical models of coastal marshes. This approach is shared by modelling studies that focus solely on hydrodynamics (e.g., Dietrich et al. (2011); Bunya et al. (2010); Defne and Ganju (2015)) and studies that include sediment transport (e.g., Castagno et al. (2018); Zang et al. (2018); Zhang et al. (2019)). In most cases, the instrumentation is installed in a few locations either in the open sea, along the coastline, or within tidal channels and creeks, as sensor installation becomes challenging on the shallow coastal marshes where boat access is difficult. Furthermore, these in-situ measurements inform about temporal variability, but provide limited spatial information across the landscape. Consequently, it is challenging to fully evaluate the quality of numerical simulations over the marsh platform, which must capture the flux of water and sediment. Despite recent progress in this direction, there remains a need to include the extensive information provided by remote sensing imagery in numerical models (Fagherazzi et al., 2020).

Over time, the introduction of more advanced sensors has improved the spatial resolution of available remote sensing imagery. Coarser resolution sensors, including the Moderate Resolution Imaging Spectroradiometer (MODIS) and the National Oceanic and Atmospheric Administration (NOAA) Advanced Very High Resolution Radiometer (AVHRR) have been used in various wetlands studies. MODIS data have the advantage of providing near daily coverage of the Earth's surface, but with a resolution of 250 or 500 m depending on the selected band (Pflugmacher et al., 2007). MODIS data timeseries have been used to monitor wetland cover (Tana et al., 2013), estuarine vegetation succession and tidal flat elevation (Zhao et al., 2009), tidal wetlands biophysical characteristics (Ghosh et al., 2016), hurricane disturbance to coastal vegetation (Wang and D'Sa, 2009), and coastal wetland biomass (Lumbierres et al., 2017). MODIS data have also been coupled with tower-based flux measurements to determine the carbon budget and the gross primary production in estuarine and coastal wetlands (Yan et al., 2008; Kang et al., 2018). Similarly to MODIS, AVHRR provides global imagery twice a day, but with a 1km resolution. AVHRR data applications vary from assessing hurricane damage in wetlands (e.g., Ill et al. (1997)) to NDVI seasonal pattern in deltaic systems (e.g., Zoffoli et al. (2008)).

Moderate spatial resolution sensors, with spatial resolution of tens of meters, provide much higher detail of the Earth's surface compared to MODIS and AVHRR. Among them, data provided by the Landsat and Sentinel sensors have been used to monitor coastal wetlands more often compared to other sensors such as the Advanced Spaceborne Thermal Emission and Reflection Radiometer (ASTER), China & Brazil Earth Resource Satellite (CBERS), Systeme Probatoire D'Observation De La Terre (SPOT 1-4) and Advanced Land Observing Satellite (ALOS) (Guo et al., 2017). Long-term Landsat timeseries have been used to detect coastal wetlands cover and area change (Cardoso et al., 2014; Couvillion et al., 2017; Kaplan and Avdan, 2017; Wang et al., 2019, 2020), wetlands vegetation classification and change (Zhang et al., 2011; Muro et al., 2016; Lopes et al., 2019; Thomas et al., 2019; Balogun et al., 2020; Zhang et al., 2022c), extreme events impact on vegetation (Rodgers

et al., 2009), coastal wetland soil vertical accretion rates (Jensen et al., 2022), and coastal wetland above-ground biomass (Tan et al., 2003; Chen et al., 2022).

Remote sensing studies have also developed algorithms to infer water quality indicators in coastal waters near wetlands (McClain and Meister, 2012). For instance, Fichot et al. (2016) used high resolution remote-sensing reflectance data from the airborne Portable Remote Sensing Spectrometer (PRISM) to derive maps of turbidity, dissolved organic carbon and chlorophyll-a in the San Francisco Bay-Delta Estuary. Jensen et al. (2019) used high resolution remote-sensing reflectance data from NASA's Airborne Visible/Infrared Imaging Spectrometer-Next Generation (AVIRIS-NG) to derive maps of total surface suspended solids in the waters of the Atchafalaya basin along the Louisiana coast (USA). They used an algorithm centered on a derivative-based partial least squares regression between measured total surface suspended solids and in-situ spectra. Other studies utilized operational satellite sensors with coarser spatial resolution and fewer spectral bands to derive sediment concentrations in coastal waters. Dorji and Fearns (2016) applied multiple algorithms to MODIS and Landsat imagery in regional waters of northern Western Australia, whereas Zhang et al. (2020a) adopted Landsat-8 and Sentinel-2 imagery to derive suspended sediment concentration in Plum Island Estuary in Massachusetts (USA).

High quality optical imagery is limited to cloud-free conditions, therefore remote sensing techniques can also rely on radar sensors to overcome this limitation (Henderson and Lewis, 2008). The ALOS Phased Array L-band Synthetic Aperture Radar (PALSAR), European Remote Sensing satellite (ERS-1), RadarSAT, Advanced Synthetic Aperture Radar (ASAR), Japanese Earth Resources Satellite 1 (JERS-1), Airborne Synthetic Aperture Radar (AIRSAR), and TerraSAR-X are some examples of radar sensors used to monitor wetlands (Guo et al., 2017). Slatton et al. (2008) showed that changes in the multi-polarization L-band AIRSAR backscatter were able to detect herbaceous vegetation in marshes, while Kwoun and Lu (2009) used SAR data over the Louisiana coastal zone to characterize seasonal variations of radar backscattering according to vegetation type. These studies show that SAR can be adopted to monitor changes in coastal wetland vegetation cover. Within the last decade repeat-pass radar interferometry from spaceborne instruments have enabled measurements of water level change within marshes (Wdowinski et al., 2008; Liao et al., 2020; Xie et al., 2013; Hong and Wdowinski, 2014). This new technology enables direct observation of large scale flow patterns that can only be observed with remote sensing.

This study presents a novel coupling between numerical modelling and high-resolution remote sensing imagery. In particular, it shows that the calibration of hydrological and sediment transport models can be performed with multiple remote sensing data. To do so, three remote sensing products from the NASA Delta-X mission are used to inform three numerical models with different spatial resolutions. The paper is structured in the following sections: an introduction to the Delta-X mission and related remote sensing products, set-up of the numerical models, results, and discussion of advantages and limitations of using these datasets with numerical models.

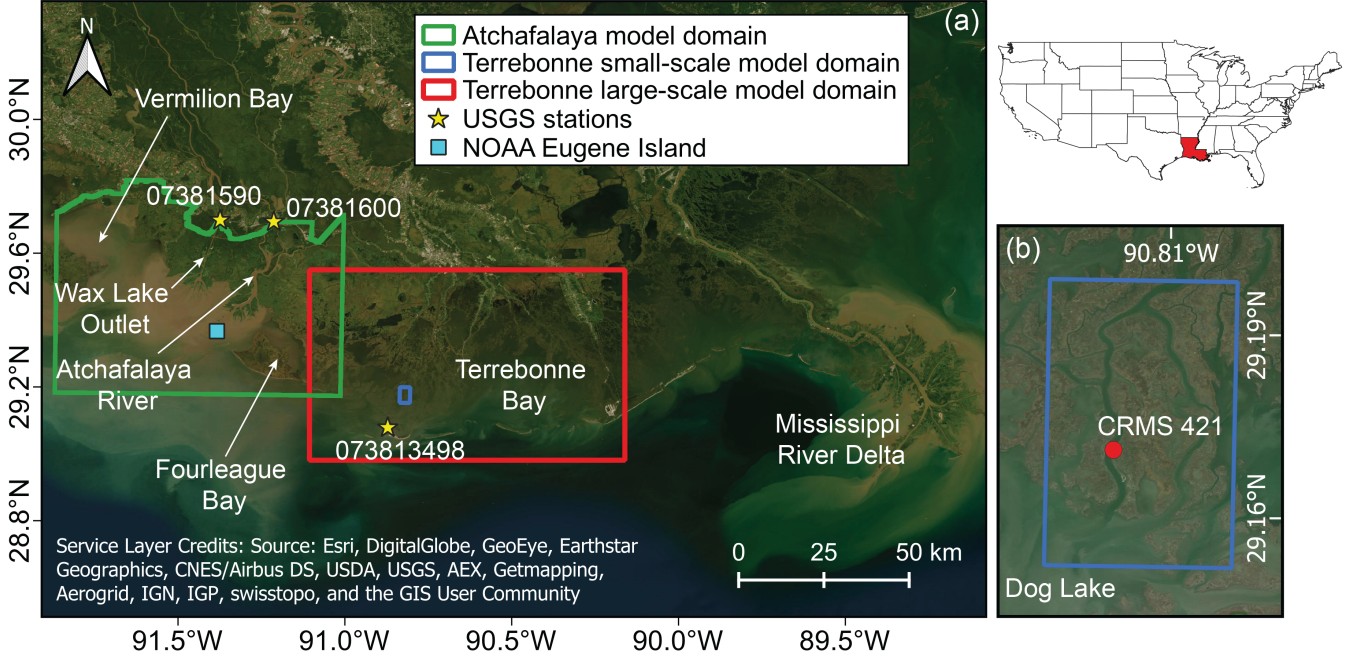

**Figure 1.** (a) The domains for the Atchafalaya and Terrebonne basin numerical models. The top left inset shows a map of the contiguous US with the state of Louisiana colored in red. (b) Inset showing a detail of the small-scale model domain in the Terrebonne basin. Sources for satellite images in Figure 1b are: Esri, DigitalGlobe, GeoEye, Earthstar Geographics, CNES/Airbus DS, USDA, AEX, Getmapping, Aerogrid, IGN, IGP, swisstopo, and the GIS User Community.

## 2    Methods

### 2.1    Study area and models domains

The Terrebonne and Atchafalaya basins are two neighboring coastal basins located along the Louisiana coast, to the west of the Mississippi River birdfoot delta (Figure 1). Among all basins in the Mississippi River Delta Plain (MRDP), the Terrebonne

wetlands have seen the highest rate of areal loss since 1932 (Couvillion et al., 2017; Jensen et al., 2022) due to lack of sediment load from inland waters. Furthermore, because of the microtidal environment, mineral soil accretion heavily relies on wind waves for bottom sediment resuspension and on storm surges to transport the sediment over the marsh platform (Cortese and Fagherazzi, 2022). The basin is characterized by a strong salinity gradient, with salt marshes dominated by *Spartina alterniflora* along the coast, and brackish and fresh marshes located in landward areas (Twilley et al., 2019).

The Atchafalaya basin is located west of Terrebonne and includes the Atchafalaya River and the Wax Lake Outlet. Both rivers mirror the seasonal pattern of the Mississippi hydrograph, with peak discharge between January and June and low discharge in September and October (Allison et al., 2000). The two rivers receive around 30% of the Mississippi River flow through the Old River Control floodgate located north of Baton Rouge (Roberts et al., 2003) and have actively growing deltas, representing

a rare instance of land gain along Louisiana's coast (Couvillion et al., 2017). For instance, the Wax Lake Delta has prograded seaward at a 270 m yr$^{-1}$ rate between 1980 and 2002 (Parker and Sequeiros, 2006). In the Atchafalaya basin nearly 80% of the wetlands are freshwater marshes and swamps due to the high freshwater discharge (Twilley et al., 2019). The distribution of the vegetation is heavily regulated by topography and hydroperiod (Bevington and Twilley, 2018).

Three numerical models were developed at different scales and calibrated with remote sensing data. Two models were set-up at the basin scale, while the third one was developed at a smaller scale. The large-scale Terrebonne model (red rectangle in Figure 1) extends longitudinally for 90 km and borders with the Atchafalaya basin to the west. The small-scale Terrebonne model is located within the salt marsh in the south-west portion of Terrebonne (blue rectangle in Figure 1) and is connected to the south with Dog Lake. The domain consists of an island surrounded by two main channels with a 100-150 m wide cross section that converge at the northern top of the island. Finally, the Atchafalaya model (green rectangle in Figure 1) extends longitudinally for 84 km and includes the two deltas, Fourleague Bay to the east, and Vermilion Bay to the west.

## 2.2  NASA Delta-X mission

Delta-X is a NASA mission funded by the Science Mission Directorate's Earth Science Division through the Earth Venture Suborbital-3 Program (Simard et al. (2022); https://deltax.jpl.nasa.gov/), which investigates how feedbacks between hydrological and ecological processes enable marshes and deltas to survive relative sea level rise. The project is focused on the two basins of the MRDP introduced above. The mission has produced airborne high-resolution remote sensing imagery and in-situ measurements that can be incorporated in hydrodynamic, sediment transport and ecological numerical models. In 2021, two field campaigns were completed, one in Spring and one in Fall in order to capture the maximum and minimum flood stages of the Mississippi river discharge.

## 2.3  UAVSAR, AirSWOT, and AVIRIS-NG

During both Delta-X campaigns, three airborne remote sensing instruments repeatedly collected data to capture the hydrological parameters and infer sediment concentration during different tidal stages.

The Uninhabited Aerial Vehicle Synthetic Aperture Radar (UAVSAR) is a fully polarimetric L-band synthetic aperture radar (SAR) with a wavelength of 23.8 cm, installed on a NASA Gulfstream-III aircraft, that can provide highly coherent rapid repeat-pass SAR acquisitions. UAVSAR data provide a measurement of water level change over the wetlands with a resolution of 6 m through repeat-pass interferometry, which allows the detection of surface displacement using multiple observations from the same viewing geometry (Rosen et al., 2006). In flooded wetlands, the water surface is detected through the double-bounce scattering mechanism from water and vegetation (Kim et al., 2009; Wdowinski et al., 2013). To separate the water surface from the emergent wetland, a water mask was generated from the interferogram Level-1 products. Figure 2a shows an example of water level change measured by UAVSAR on the wetlands in western Terrebonne between 17:13 and 17:44 (GMT) on 06 April 2021. A phase unwrapping algorithm is employed to convert interferometric phase change $\Delta\phi$ to change in elevation $\Delta z$ (Oliver-Cabrera et al., 2021). Here, Level-3 UAVSAR data acquired in Terrebonne are used (Jones et al., 2022).

AirSWOT is an airborne Ka-band synthetic aperture radar with a wavelength of 0.84 cm flown on a Beechcraft King Air B200, that measures water surface elevation and water surface slope in open waters with uncertainty below 0.3 cm/km. Air-SWOT uses cross-track interferometry to measure the elevation and combines it with along-track interferometry to correct for the bias due to the water motion (Goldstein and Zebker, 1987). To separate land from water surfaces, the same UAVSAR water mask was used for AirSWOT. More details on the application of AirSWOT are reported by Denbina et al. (2019). Figure 2b shows an example of water surface elevation in western Terrebonne acquired along a flight line on 05 April 2021 at 22:22 GMT. Here, Level-2 AirSWOT geocoded water surface elevation data in Terrebonne and Wax Lake (Denbina et al., 2022) are used. Water surface elevation data was validated in both Delta-X campaigns using in-situ gauges and a root mean squared error of 9 cm was found when data were averaged on a 1 km$^2$ area.

The Airborne Visible/Infrared Imaging Spectrometer-Next Generation (AVIRIS-NG) is a high-resolution imaging spectrometer that measures radiance for 432 bands at 5-nm spectral sampling between 380 and 2510 nm (Hamlin et al., 2011). The calibrated radiance measurements from AVIRIS-NG were atmospherically corrected to produce spectral remote-sensing reflectance ($R_{rs}(\lambda)$) of the water, and surface reflectance of the land. Local empirical algorithms derived using in-situ measurements were used to derive TSS concentration from the $R_{rs}(\lambda)$ in the visible/near-infrared region and generate maps of TSS from the AVIRIS-NG imagery (Gao et al., 1993; Bue et al., 2015; Jensen et al., 2019). In-situ samples were collected in both Terrebonne and Atchafalaya basins during the Delta-X 2021 Spring and Fall campaigns in order to capture high and low flow conditions. The algorithm to retrieve TSS from AVIRIS-NG performed well (Median Absolute Percent Difference 13.7% and Median bias 6.71 mg/l) across a wide range of TSS concentrations (0.1–154.5 mg/l) (Fichot and Harringmeyer, 2021, 2022). AVIRIS-NG images were also used to produce maps of vegetation structure (Jensen et al., 2021). Figure 2c shows an example of total suspended solids (TSS) concentration maps derived from AVIRIS-NG over the Wax Lake Outlet acquired on 02 April 2021 at 20:31 GMT. Here, the AVIRIS-NG Level3-derived TSS data in Terrebonne and Wax Lake are used (Fichot and Harringmeyer, 2022).

## 2.4 Large-scale Terrebonne model set-up

The Delft3D model was used to simulate water levels and sediment transport during the Delta-X 2021 Spring campaign. The FLOW module (Lesser et al., 2004) was coupled with SWAN (Simulating Waves Nearshore, e.g. Holthuijsen et al. (1993); Booij and Holthuijsen (1987)) to account for wave resuspension of bottom sediments. The model ran from 25 March 2021 to 18 April 2021 with an additional 5 days as spin-up period. The domain (Figure 1) consisted of 1139 × 686 cells with a 90 × 90 m resolution. Bathymetric data were referenced to the NAVD88 vertical datum and are available from NOAA (Love et al., 2010). Boundary conditions were water levels imposed at the south boundary in the Gulf of Mexico. Water level data were taken from the United States Geological Survey (USGS) station at Caillou Bay SW of Cocodrie (ID:073813498). The water levels at the boundary were adjusted to reproduce the correct water levels at the USGS station location by shifting the phase and correcting the amplitude to account for signal damping. Data of wind speed and direction were taken from the same USGS station and applied homogeneously on the entire domain with an hourly time resolution. Two sediment types composing the bottom were considered: sand (non-cohesive fraction) and mud (cohesive fraction). The initial sediment distribution was

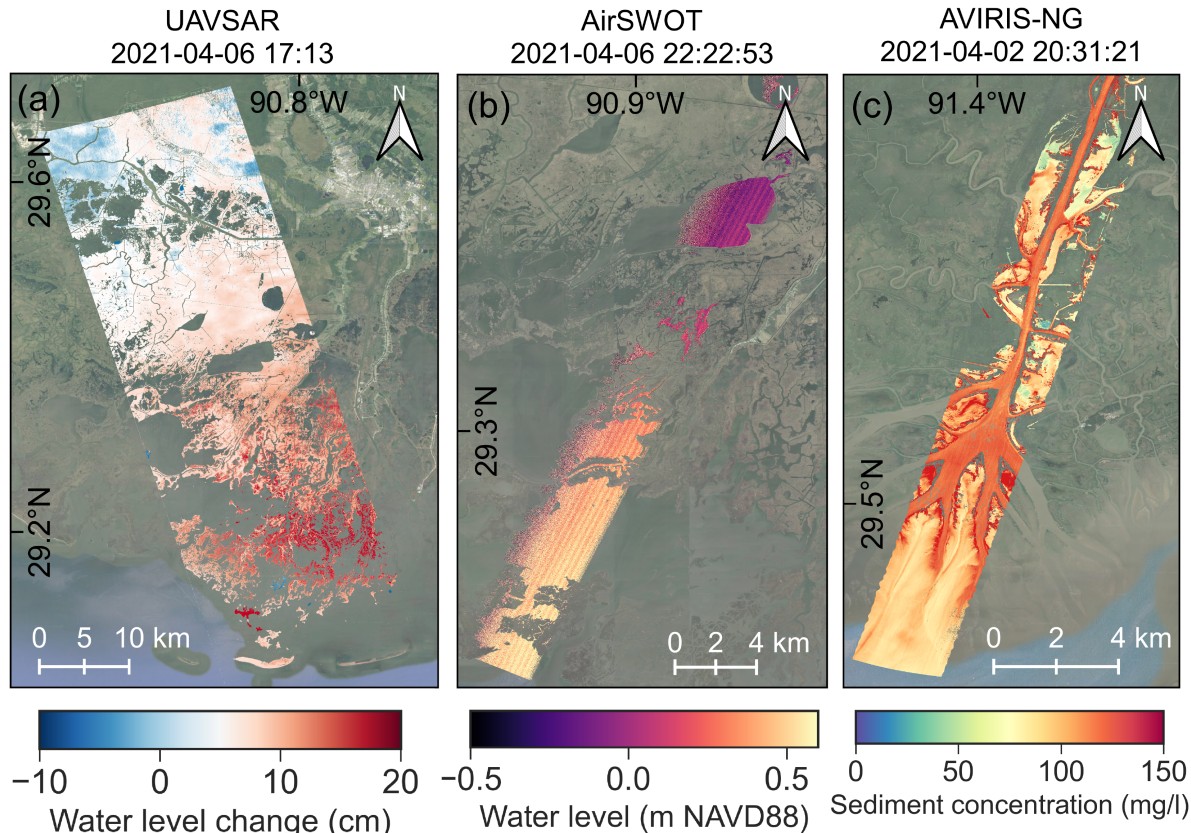

**Figure 2.** Example of Delta-X remote sensing products. At the top of each map the acquisition time is reported in GMT. (a) Water level change on the marsh from UAVSAR. (b) Water elevation from AirSWOT. (c) Total suspended solids (TSS) concentration derived from AVIRIS-NG. Refer to Figure 3 for the spatial relationship. Basemap credits: ©Google Satellite.

derived from the usSEABED database (Williams et al., 2006). Field measurements were interpolated to derive the fraction of mud and sand in each cell.

The model was coupled with AirSWOT to calibrate bed roughness expressed as Chezy coefficient. Three different friction categories were considered: open sea/ocean, tidal channels/lakes/bays, and marsh platform. Since the AirSWOT campaign covers the water area, the calibration of the Chezy coefficient was focused only on the tidal channels/lakes/bays. Values of 65

and 35 $m^{1/2}s^{-1}$ were set for ocean and marsh platform respectively, whereas 45, 55, and 65 $m^{1/2}s^{-1}$ were evaluated for the tidal channels/lakes/bays. The selected values fall within the range adopted by several modelling studies of coastal marshes and deltas (e.g. Edmonds and Slingerland (2010); Nardin et al. (2013); Stark et al. (2015); Zhang et al. (2019)).

The model was coupled with AVIRIS-NG to calibrate sediment parameters. Sand density and median diameter ($D_{50}$) were set to be constant at 2650 $kg/m^3$ and 0.14 mm, respectively. Note that in the case of non-cohesive particles, Delft3D does not

require to specify a value for the settling velocity, since it is directly computed from the median diameter and density using

the Van Rijn et al. (1993) approach. Thus, the calibration of the parameters refers to the properties of the mud fraction. The default transport equation of Van Rijn (2007) was used for sand. In Delft3D, the cohesive sediment is defined by density, settling velocity, critical shear stress for erosion/sedimentation, and erosion parameter. In this case, the settling velocity, $w_s$, and critical shear stress for erosion, $\tau_{cr,e}$, were calibrated. In particular, the last parameter is the threshold above which the applied shear stress is able to entrain bottom sediment. The default Delft3D sediment transport formulation of Partheniades-Krone (Partheniades, 1965) was used for the cohesive fraction. In total, five possible values for $w_s$ (0.1, 0.175, 0.25, 0.325, 0.4 mm/s) and five possible values for $\tau_{cr,e}$ (0.05, 0.075, 0.1, 0.125, 0.15 Pa) were tested. Mud density was set at 1600 kg/m$^3$ (Liu et al., 2018), while the erosion parameter was fixed at $1 \cdot 10^{-5}$ kg m$^{-2}$s$^{-1}$ consistently to previous modelling studies of US coastal bays (Ganju and Schoellhamer, 2010; Wiberg et al., 2015).

## 2.5 Small-scale Terrebonne model set-up

Delft3D was utilized to simulate the hydrodynamics in one of the Delta-X intensive field study sites near station 421 of the Coastwide Reference Monitoring System (CRMS) network (blue rectangle in Figure 1) from 25 March 2021 to 18 April 2021. An additional interval of time of 5 days was used as a spin-up period. The numerical grid had a resolution of 10 m and consists of 300 × 520 cells. Bathymetric information comes from LiDAR, and is given with respect to NAVD88 (Denbina et al., 2020). Water levels were imposed as boundary conditions at the south and north boundaries. Water levels at both boundaries were extracted from the large-scale model in the respective locations. Bottom friction was imposed in terms of Chezy coefficients: 55 m$^{1/2}$s$^{-1}$ for the channels and 35 m$^{1/2}$s$^{-1}$ for the marsh platform. A depth threshold of 1 cm was set for the wetting-drying scheme.

To correct marsh topography, the methodology proposed by Zhang et al. (2022a) was followed, in which errors in elevation were corrected by comparing modelled water-level changes with those observed via UAVSAR (see Figure S1). In this procedure, the first simulation is run using the original topography and the difference between modelled and observed water level change is computed in each cell. If the modelled water level change is larger than UAVSAR, the elevation of the cell is increased. In the opposite case, the elevation is decreased. The updated topography is used in the subsequent simulation. The procedure is run with the updated topography iteratively until the minimum RMSE is reached. As suggested by Zhang et al. (2022a), changes in elevation in each iteration were small and gradual to allow the system to adjust. The method is considered successful only if converges to one solution and the final topography is realistic.

## 2.6 Atchafalaya model set-up

Similar to the large-scale model in Terrebonne, a Delft3D-FLOW model was developed to simulate water levels and sediment transport in the Atchafalaya basin (Figure 1) during the Delta-X 2021 Spring campaign. The model had 927 × 787 cells with a 90x90 m resolution. The model ran from 15 March 2021 to 25 April 2021. Bathymetric data were referenced to NAVD88 (Denbina et al., 2020). The model had a total of three imposed boundary conditions. Water discharge from the USGS station at Calumet (ID: 07381590) was imposed for the Wax Lake Outlet, while water discharge from the USGS station at Morgan City (ID: 07381600) was imposed for the Atchafalaya River. The two stations also provided suspended sediment concentration data

**Table 1.** Calibrated parameters and relative remote sensing data used for each model.

| Model | Calibrated parameter | Remote sensing data used |
|---|---|---|
| Large scale Terrebonne | $C_h$ / $w_s$, $\tau_{cr,e}$ | AirSWOT/AVIRIS-NG |
| Small scale Terrebonne | Bathymetry | UAVSAR |
| Atchafalaya | $C_h$ / $\tau_{cr,e}$ | AirSWOT/AVIRIS-NG |

that were imposed as boundary conditions for the sediment transport model. Water levels from the NOAA station at Eugene Island (ID: 8764314) were used at the oceanic boundary. Since the oceanic boundary is located further offshore, we adjusted the signal using the same procedure described in the Terrebonne large scale model. Hourly wind speed and direction data were retrieved from the same station and applied uniformly over the domain.

Three categories were considered for the bed friction calibration with AirSWOT. The Chezy coefficient was fixed at 35 $\mathrm{m}^{1/2}\mathrm{s}^{-1}$ for the marsh, and 8 $\mathrm{m}^{1/2}\mathrm{s}^{-1}$ for the forest (Zhang et al., 2022a). The evaluated values for open water friction were 45, 55, and 65 $\mathrm{m}^{1/2}\mathrm{s}^{-1}$.

For sediment transport, three sediment classes were considered: sand, silt, and mud. The initial sediment distribution was derived from the usSEABED database and set uniform at 22% sand, 39% silt, and 39% clay. Settling velocities for silt and clay were fixed at 1 mm/s and 0.5 mm/s, respectively, while the median diameter of sand was set at 0.1 mm. Due to the limited number of samples in the area, a bed spin-up process of one year was run with a morphological speeding factor of 50 to reach bottom equilibrium. During this process, the bed level was kept fixed, and the bed fractions changed to adapt to the hydrodynamics (see similar approaches in van der Wegen et al. (2011) and Zhang et al. (2020b)). The bed composition after this spin-up process became realistic and the coarser fractions appeared in areas with strong bottom shear stress. AVIRIS-NG sediment concentration maps were used to calibrate the critical shear stress for erosion. Evaluated values were 0.025, 0.03, 0.04. 0.05, 0.1 Pa for clay, and 0.05, 0.1, 0.2 Pa for silt. Performance of the sediment transport model was also evaluated with and without waves, which were computed with SWAN.

## 2.7 Coupling between imagery and numerical model

Table 1 summarizes the calibrated parameters for each model and the corresponding remote sensing data used, while Figure 3 shows the chosen flight lines of the three instruments.

In all three models, the coupling between remote sensing images and model allowed to tune parameters to best match numerical outputs and the observations. Water levels and sediment concentration from the numerical models were extracted using the spatial extent of the chosen flight lines and overlapped to enable comparisons. The comparison was first performed visually, in order to identify evident discrepancies. Then, a pixel-by-pixel comparison was performed to better quantify model performance. For the Terrebonne large-scale model and the Atchafalaya model, because of the 90 m mesh resolution, both AirSWOT and AVIRIS-NG resolution was lowered by averaging values within a 90 m mesh element. The evaluation of the models was performed by computing the error as the difference between the measurement and the modelled values, with a

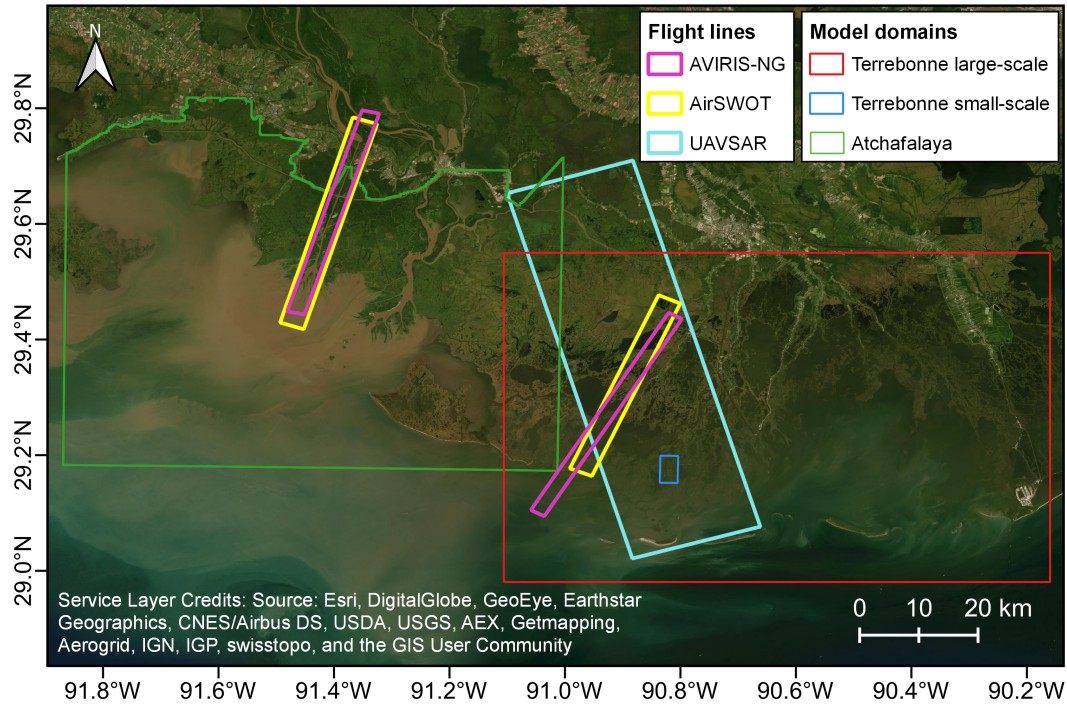

**Figure 3.** Extension of the flight lines used in this study. The extension of the numerical models is also reported with thinner lines.

positive error indicating underestimation. The root mean square error (RMSE) was adopted as a global metric to compare the overall performance of each simulation. Parameter values providing the lowest RMSE were identified as the optimal ones.

In the large-scale Terrebonne model, a second calibration of the Chezy coefficient and sediment properties was performed using only in-situ observations. Simulated water levels during the Spring campaign were compared with timeseries of 13 tidal gauges within the CRMS network (Figure S3). The RMSE and Nash-Sutcilfe Model Efficiency (ME) (Allen et al., 2007) were used to evaluate model performance for the different friction coefficients. The validation with timeseries allowed to evaluate the temporal coherence of the results that cannot be captured by remote sensing. The two calibrations were validated in the Fall campaign by comparing water levels. In the same model, a calibration of sediment parameters using in-situ measurements was carried out. Measured TSS concentrations from Fichot et al. (2022) were compared to simulation results (see Figure S5). Finally, the two calibrations were validated with in-situ TSS data collected during the Fall campaign (Fichot et al., 2022). RMSE was used to compare model results.

To validate the topographic correction in the small-scale model, the original and calibrated topography were compared with Real Time Kinematics elevation measurements collected during the Delta-X Spring 2021 campaign (Twilley and Rovai, 2022) and the site 421 elevation provided by the CRMS.

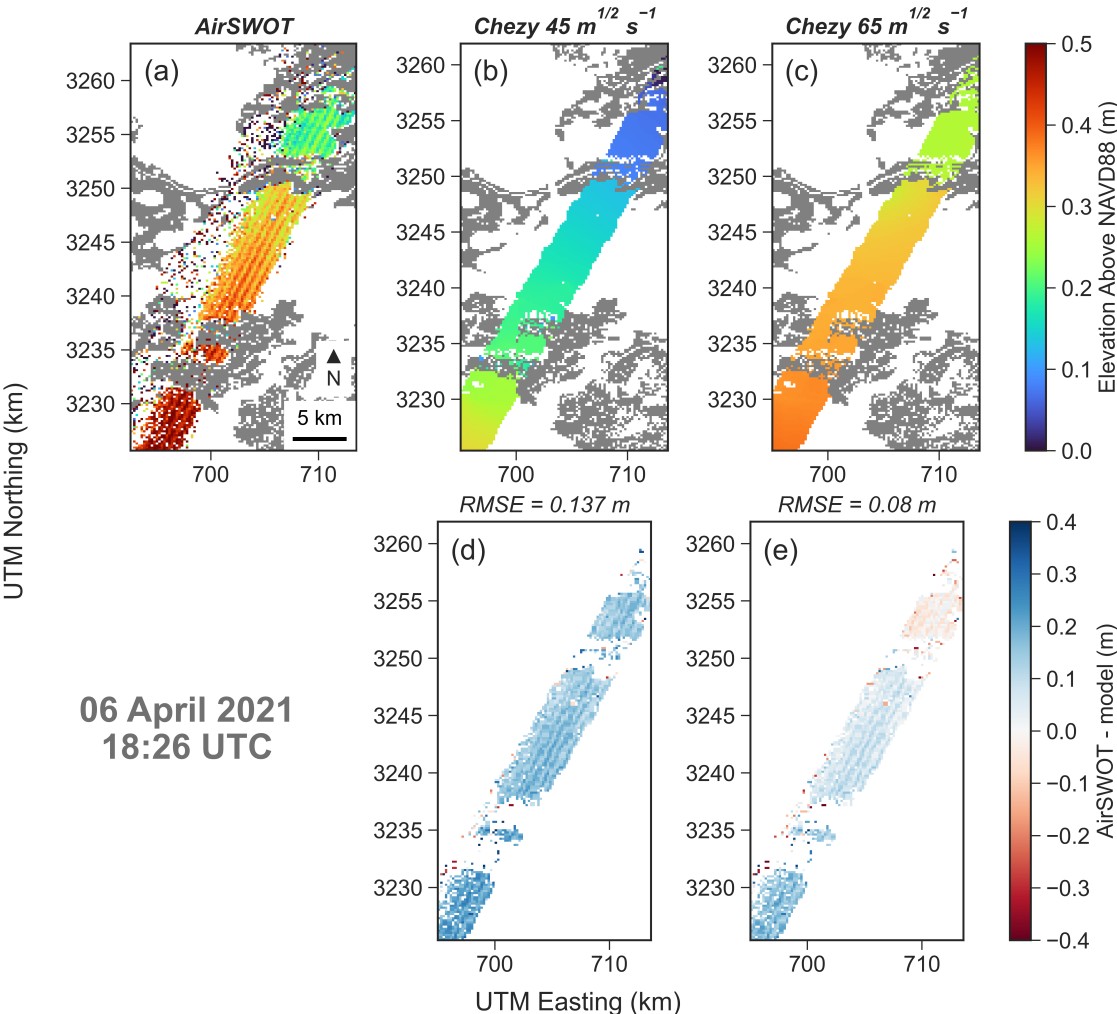

**Figure 4.** Comparison between measured water levels by AirSWOT and modelled water levels within the large-scale Terrebonne model. From left to right are presented AirSWOT elevation, modelled elevation with $C_h$ = 45 $\mathrm{m}^{1/2}\mathrm{s}^{-1}$ and modelled elevation with $C_h$ = 65 $\mathrm{m}^{1/2}\mathrm{s}^{-1}$. On the bottom row the difference between AirSWOT and the model results are displayed with the overall RMSE.

## 3 Results

### 3.1 Coupling with AirSWOT

For the Terrebonne large scale model, results of simulations with Chezy values of 65 and 45 $\mathrm{m}^{1/2}\mathrm{s}^{-1}$ are shown for one acquisition (Figure 4). The comparisons for four acquisitions of the same flight line at different times during the tidal cycle are reported in Figure S2. The flight line has south-west to north-east direction. On the south, it crosses a channel with a 500-700 m cross-section and one of the major lakes in Terrebonne (Caillou Lake). On the north, it intersects a portion of other lakes

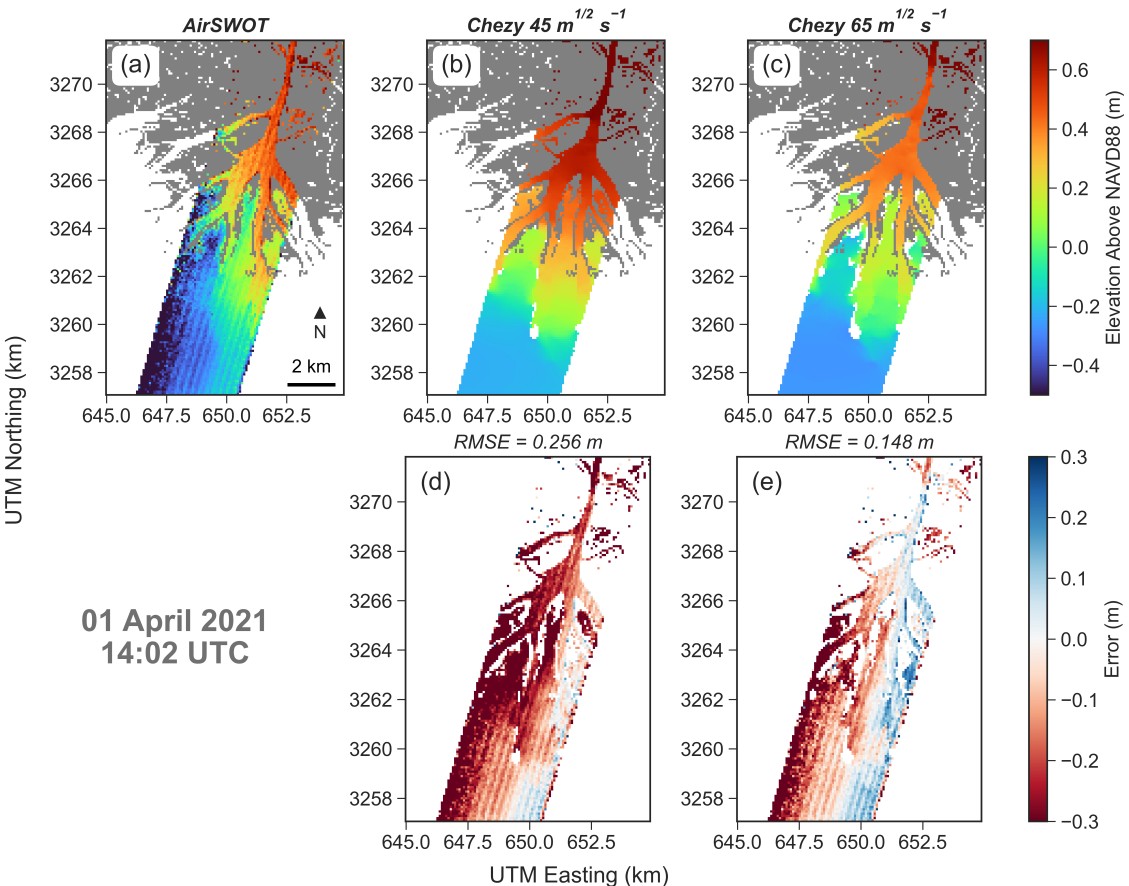

**Figure 5.** Comparison between measured water levels by AirSWOT and modelled water levels in Atchafalaya Bay (with focus on the Wax Lake Delta). From left to right are presented AirSWOT elevation, modelled elevation with $C_h$ = 45 m$^{1/2}$s$^{-1}$ and modelled elevation with $C_h$ = 65 m$^{1/2}$s$^{-1}$.

and small channels. A Chezy coefficient of 65 m$^{1/2}$s$^{-1}$ returns a better match with the AirSWOT observations excepth for the third acquisition on 06 April 2010 UTC (Figure S2C), were slightly better results were found for a Chezy coefficient of 45 m$^{1/2}$s$^{-1}$. The model displays better agreement in the south section of the flight line, whereas the discrepancy between remote sensing data and model results tends to increase in the north section. The use of a friction coefficient of 65 m$^{1/2}$s$^{-1}$ allows to better represent the water levels in the northern section of the flight line.

Figures S5a and S5b show the results of the calibration using timeseries of stationary gauges. The calibration shows similar results, with $C_h$ = 45 m$^{1/2}$s$^{-1}$ providing the better RMSE. The validation of the friction coefficient during the Fall campaign (Figures S5c and S5d) shows better performance for $C_h$ = 65 m$^{1/2}$s$^{-1}$, with a RMSE of 0.047 m compared to the 0.066 m for the 45 m$^{1/2}$s$^{-1}$ case.

**Table 2.** Parameters for the large-scale models calibrated with AirSWOT and AVIRIS-NG data.

| Parameter | Terrebonne | Atchafalaya |
|---|---|---|
| $C_h$ (m$^{1/2}$s$^{-1}$) | 65 | 65 |
| Mud: $w_s$ (mm/s) | 0.25 | - |
| Mud: $\tau_{cr,e}$ (Pa) | 0.1 | - |
| Clay: $\tau_{cr,1}$ (Pa) | - | 0.03 |
| Silt: $\tau_{cr,2}$ (Pa) | - | 0.1 |

The calibrated value of 65 m$^{1/2}$s$^{-1}$ for the Chezy coefficient is comparable to Manning values for the area. If we consider an average modelled water depth of 1.34 m during the AirSWOT acquisitions and use the water depth as an approximation for hydraulic radius, it can be estimated a correspondent Manning coefficient n = H$^{1/6}C_h^{-1}$ (Limerinos, 1970). The equation yields n = 0.02 m$^{1/3}$s$^{-1}$ which is within the range of values suggested by the 1992 NLCD and LA-GAP classification for the open water class (Bunya et al., 2010).

Atchafalaya model results with Chezy values of 45 m$^{1/2}$s$^{-1}$ and 65 m$^{1/2}$s$^{-1}$ (Figure 5) are shown for the Wax Lake Delta. A Chezy coefficient of 65 m$^{1/2}$s$^{-1}$ drastically improves model performance compared to the 45 m$^{1/2}$s$^{-1}$ case, with a reduction of the RMSE from 25.6 cm to 14.3 cm. In particular, the reduction of bottom friction allows the model to better represent water elevations in the main channel, but especially in the vicinity of the distributary channels of the delta where the model with 45 m$^{1/2}$s$^{-1}$ tends to overestimate water elevations. Furthermore, the 65 m$^{1/2}$s$^{-1}$ simulation detects the west-to-east surface

slope.

### 3.2   Coupling with UAVSAR

Data of water-level changes measured during the 12 April 2021 UAVSAR flight were used to correct marsh topography. The campaign took place between $t_1$=19:29 and $t_2$= 22:59 (UTC time). The data collected in this temporal window offer a synoptic view of water-level change across the marsh landscape during falling tides. Since water level change is computed as the

difference between the measurements at $t_2$ and $t_1$, the negative values in Figure 6a confirms that the waters on the marsh are receding. Results from the first run (Figures 6b and 6d) highlight two critical areas. First, the model overestimates the water level change in the southern area, indicating an error in the marsh elevation derived from LiDAR data. The opposite is occurring in the northern area, where waters are found to recede too slowly and consequently the water level change is underestimated. Thus, the original topography (Figure 6f) was modified with the described step-wise approach, and after four iterations the new

topography (Figure 6g) increased model performance (Figure 6c), with RMSE decreasing from 4.9 to 3.9 cm. During each step, model performance increased and after four iterations, changes in topography and RMSE were negligible. On average the total elevation change was -0.013 m, indicating the general need to lower the elevation. The topography modifications (Figure 6h) well reproduced water level changes (Figure 6a), particularly in the critical areas previously highlighted.

Validation results are reported in Figure S5. Overall, the correction based on UAVSAR decreases the elevation error from 18.94 cm to 9.78 cm. Despite the general improvement, for some specific points the correction increased the error. In particular, the bathymetry was strongly deepened at three locations.

## 3.3 Coupling with AVIRIS-NG

For the Terrebonne model, the AVIRIS-NG acquisition taken on 05 April 2021 at 19:57:00 GMT (Figures 7 and 8) is considered. The flight line crosses the open coast and Caillou Lake from south-west to north-east. Results show that sediment concentrations tend to decrease as the critical shear stress increases (Figure 7). For $\tau_{cr,e}$ = 0.05 Pa, the models tends to overestimate the sediment concentrations. The best agreement (lowest RMSE) occurs at value 0.1 Pa. For $\tau_{cr,e}$ = 0.15 Pa, the error increases again with an underestimation of sediment concentration. Figure 8 shows the effect of settling velocity. When $w_s$ = 0.1 mm/s the sediment concentration is grossly overestimated. With higher values of settling velocity, the comparison improves, until it reaches the lowest RMSE of 34.79 mg/L for $w_s$ = 0.25 mm/s. For higher values, the agreement declines, and the model tends to underestimate sediment concentrations. The flight line also crosses a meander of a large channel (center-right position). Here, the model tends to always overestimate sediment concentration and the error decreases as critical shear stress and settling velocity increases. Overall, the model performs better in the open sections of the flight line, while at the extremities, performance declines. The combination that provides the best comparison with measurements is a critical shear stress of 0.1 Pa and a sediment settling velocity of 0.25 mm/s.

Settling velocity calibrated using only in-situ measurements was found to be 0.325 mm/s, different from the value obtained from the remote sensing images (Figure S7). The optimal critical shear stress was identical (0.1 Pa). This combination provided the lowest RMSE of 25.26 mg/L. During the Fall campaign, when the two calibrations were compared, the calibration performed using AVIRIS-NG provided better agreement with in-situ measurements than the calibration with in-situ measurements only (Figure S8).

For the Wax Lake model, the acquisition taken on 02 April 2021 at 19:59:22 GMT is considered (Figures 9 and 10). A lower critical shear stress generates higher sediment concentrations, consistent with the Terrebonne model. In particular, the best match is obtained for $\tau_{cr,1}$ = 0.03 Pa (clay) and $\tau_{cr,2}$ = 0.1 Pa (silt), with a RMSE of 35.88 mg/L (Figure 10g). In this case, results improve when waves are included in the simulation. This is evident in the shallow areas near the delta, suggesting that wave resuspension must be included despite the system is river-dominated. In the other two cases the model tends to underestimate ($\tau_{cr,1}$ = 0.1 Pa and $\tau_{cr,2}$ = 0.2 Pa) and overestimate ($\tau_{cr,1}$ = 0.025 Pa and $\tau_{cr,2}$ = 0.05 Pa) sediment concentrations.

## 4 Discussion

The combination of AirSWOT, UAVSAR, and AVIRIS-NG enables calibration and evaluation of different parameters of the numerical model at spatial scales that in-situ point measurements cannot capture. AirSWOT and UAVSAR data can greatly improve hydrodynamic models. In particular, their combination enables evaluation of the goodness of computed water fluxes in

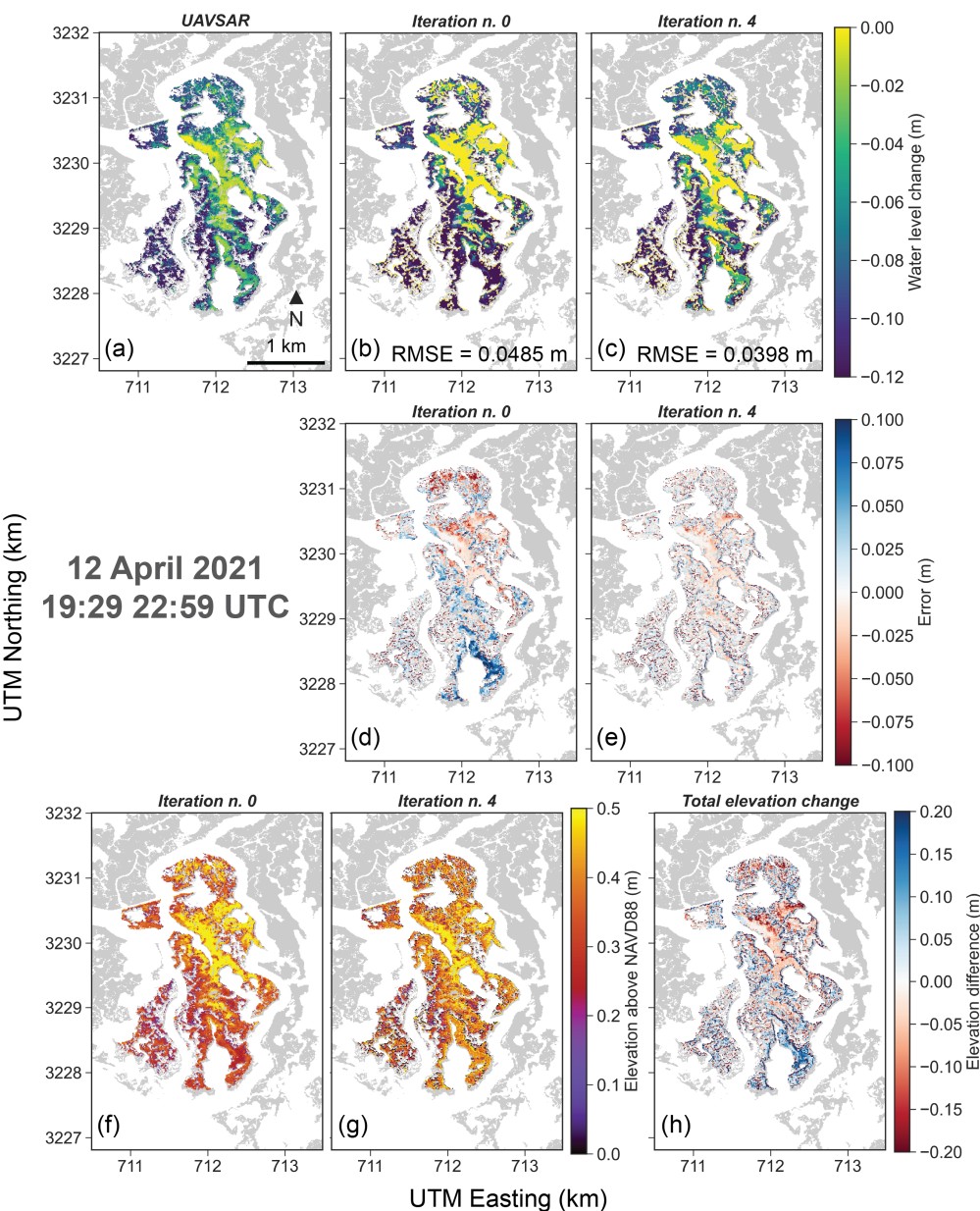

**Figure 6.** Comparison between (a) water-level changes in vegetated areas measured by UAVSAR during the April 12 2021 UAVSAR flight between 19:29 and 22:59 (UTC time), and water-level changes obtained via numerical modelling by using: (b) the initial marsh topography, and (c) the corrected topography after four iterations. Subplots (d) and (e) show the model error at the first and fourth iteration respectively. Comparison between (f) initial marsh topography and (g) marsh topography after four iterations of the correction method. Subplot (h) shows the total implemented changes in bed elevation.

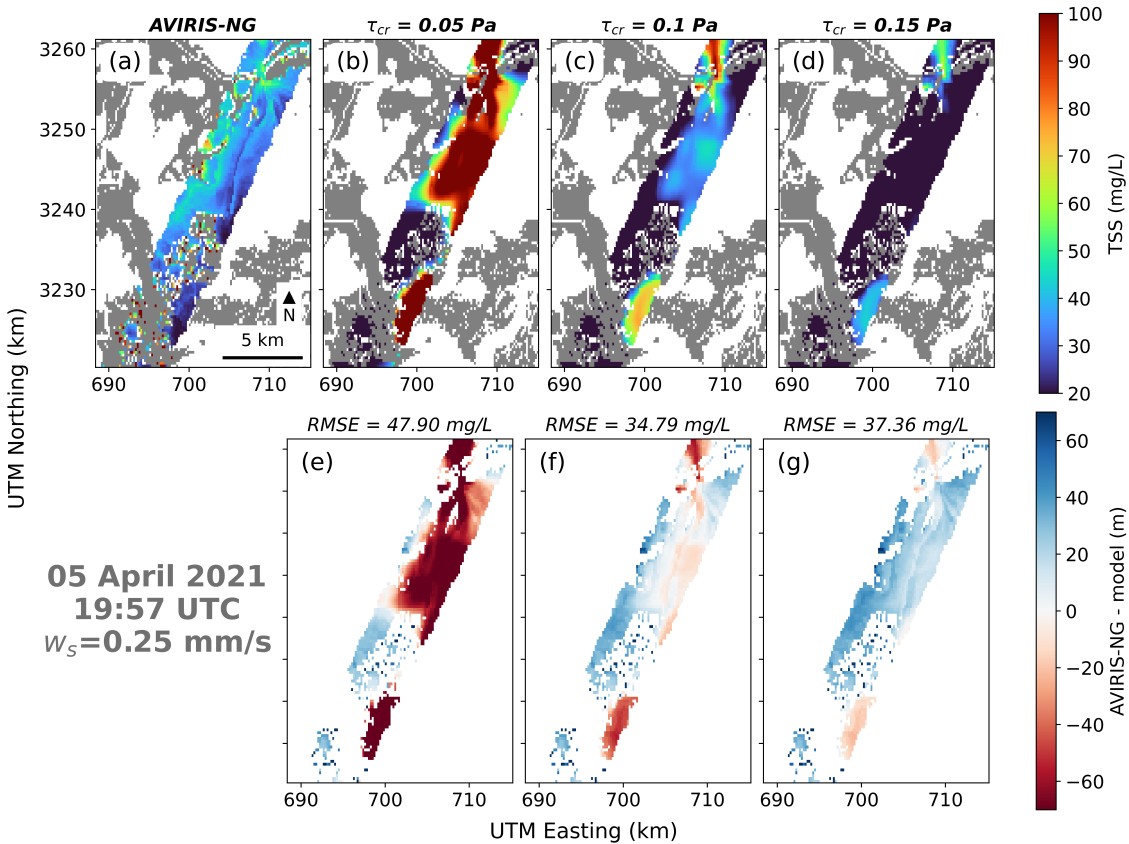

**Figure 7.** Comparison between AVIRIS-NG and the modelled sediment concentration. The settling velocity is fixed at 0.25 mm/s and the critical shear stress is changed. The first map on the top row shows the sediment concentration derived from AVIRIS-NG, while the next three maps show the modelled sediment concentration for different critical shear stress. The bottom rows show the error between measurements and model. At the top of each error map the RMSE is indicated. The sediment concentration colorbar is fixed between the minimum and maximum value of the measured sediment concentration, while the error colorbar is fixed between the overall minimum and maximum computed error.

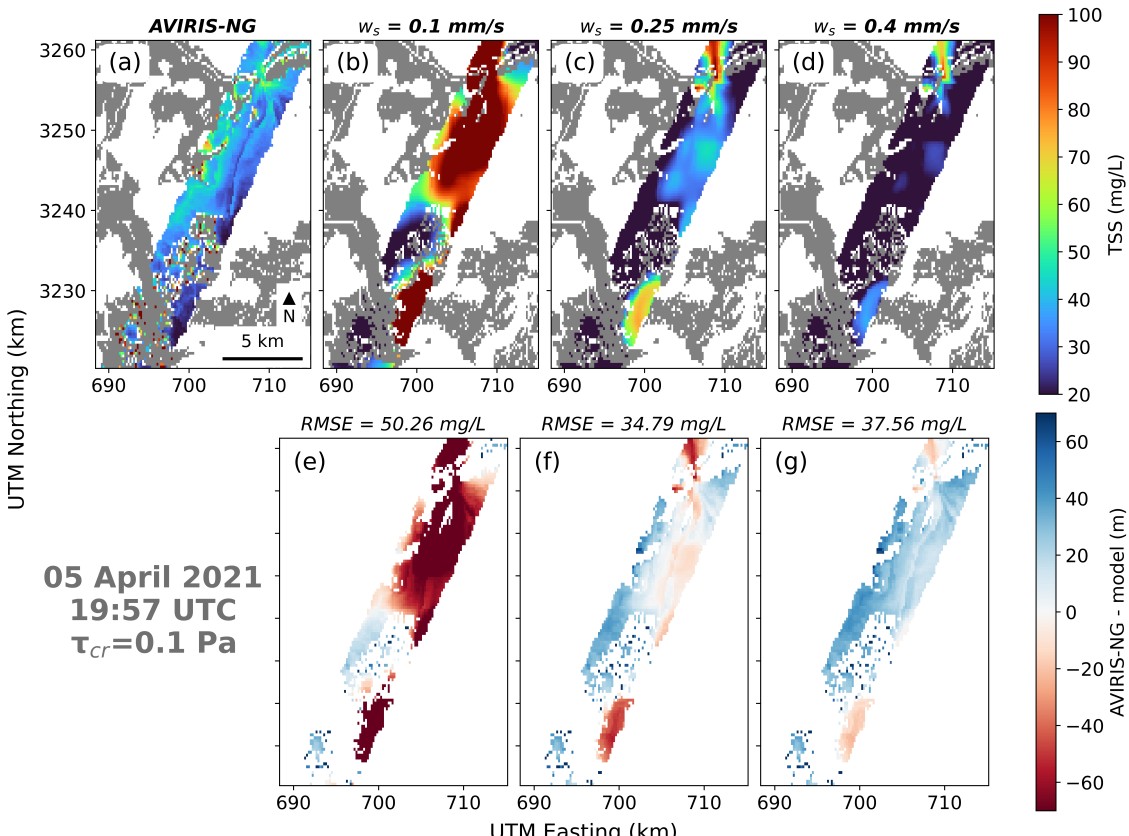

**Figure 8.** Comparison between AVIRIS-NG and the modelled sediment concentration. The critical shear stress is fixed at 0.1 Pa and the settling velocity is changed. The first map on the top row shows the sediment concentration derived from AVIRIS-NG, while the next three maps show the modelled sediment concentration for different settling velocities. The bottom rows show the error between measurements and model. At the top of each error map the RMSE is indicated.

bays, tidal channels, and on the marsh platform. AVIRIS-NG-derived sediment concentration enables evaluation of the models' ability to reproduce sediment resuspension and transport.

## 4.1 AirSWOT

The comparison between hydrodynamic models and remote sensing imagery shows that the model can reproduce the water levels in the southern half of the flight line. The results are comparable to other modeling studies at the same location and in similar intertidal areas along the US Atlantic coast. For instance, Mariotti et al. (2010) and Palazzoli et al. (2020) developed models for the shallow coastal bays of the Virginia Coastal Reserve and obtained RMSE of 0.07-0.11 m and 0.26 m respectively. In Louisiana, Freeman et al. (2015) modelled the impact of Hurricane Rita in one of the major coastal lakes in Terrebonne Bay

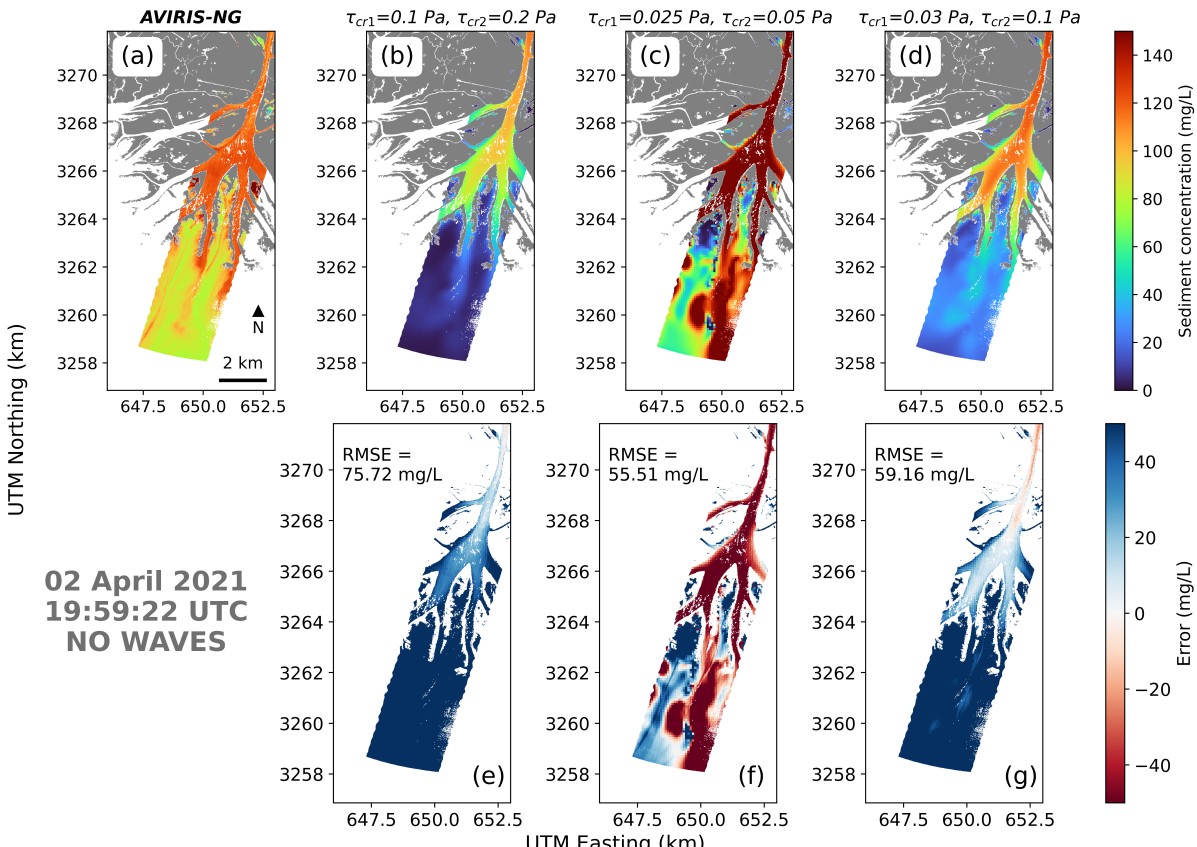

**Figure 9.** Comparison between AVIRIS-NG and the modelled sediment concentration in Wax Lake Delta without waves. Scenarios show model results with different critical shear stresses for clay ($\tau_{cr,1}$) and silt ($\tau_{cr,2}$).

obtaining a RMSE of 0.1 m, whereas Ou et al. (2020) developed a salinity model in Barataria Bay and found a RMSE between
0.05 and 0.14 m for water levels.

In the northern part of the large-scale Terrebonne Bay model, water levels tend to be regularly underestimated especially in the most areas further north not shown in Figures 4 and Figure S2. AirSWOT data highlight zones where model performance needs to be addressed. Errors can be related to uncalibrated bed friction coefficients. In some cases, the model underestimates the water levels by 0.3 m, which represents a significant error since the tidal range is around 0.4 m (Georgiou et al., 2005). The
issue could be addressed by increasing the Chezy coefficient. However for such high discrepancy between water levels, this would likely yield to large and unrealistic values. In this case, there is a hydraulic connectivity problem, namely, the model is not able to fully propagate the tides in the northern areas. The limitation lies in the coarse resolution of the mesh (90x90 m), which cannot capture the intricate network of narrow channels that connects the upper and lower portions of the domain. One way to address the issue is to carve and enlarge the channels until there is a satisfactory match between model and remote

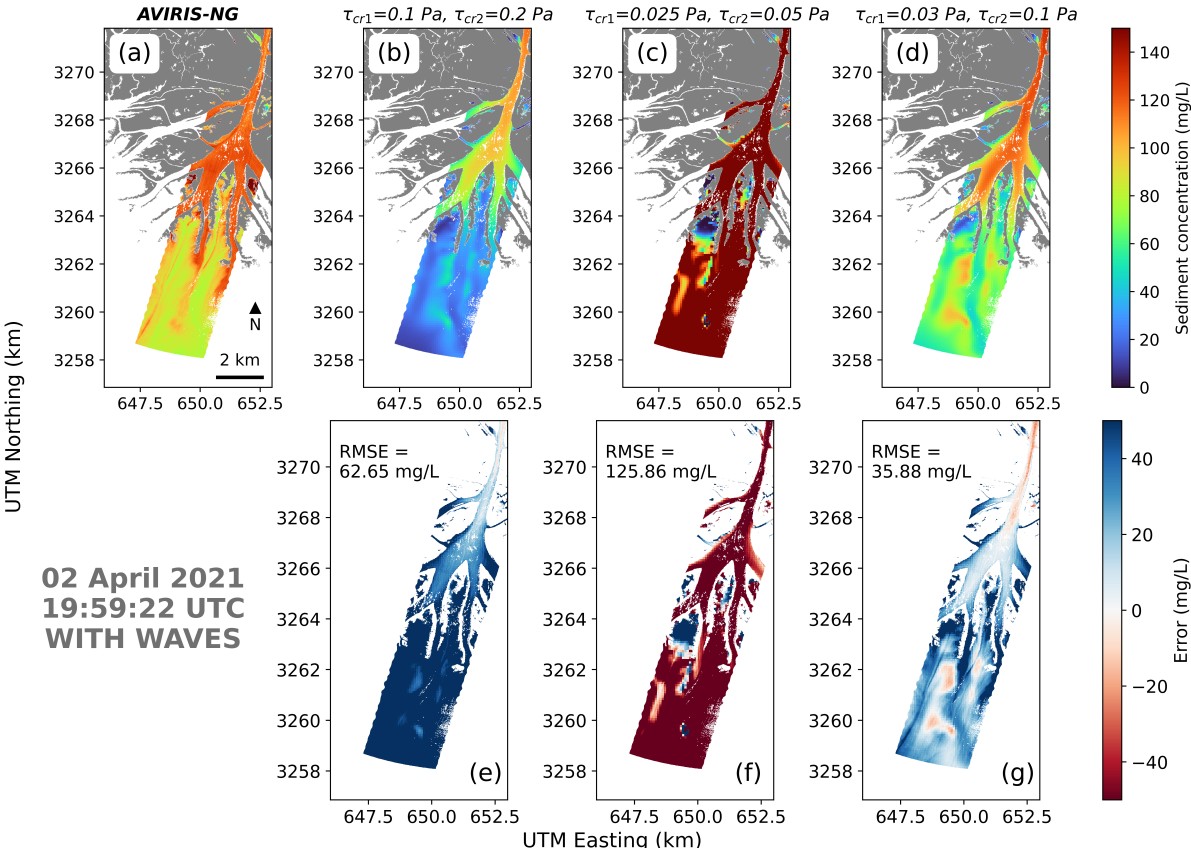

**Figure 10.** Comparison between AVIRIS-NG and the modelled sediment concentrations in Wax Lake Delta with the effect of waves. Scenarios show model results with different critical shear stresses for clay ($\tau_{cr,1}$) and silt ($\tau_{cr,2}$).

sensing data (Zhang et al., 2022b). It has to be noted that the AirSWOT flight lines presents a large error at the edges (blue areas in Figure 5a) because the incident angle of the outer swath mode is within 4 and 25 degrees (Denbina et al., 2019).

In the Atchafalaya model, similar errors are observed. Water elevations in areas located in the wetlands north of Wax Lake Delta tend to be overestimated. The reason can be similarly explained as in the Terrebonne model: the low spatial resolution does not correctly represent the narrow channels. However, the model well replicated the water levels in the delta due to the wide cross section of the main distributaries (between 200-500 m).

It is also worth noting that the correct boundary conditions must be imposed before calibration with the remote sensing imagery. The boundary condition in the large-scale model is the water level signal from the Gulf of Mexico, which was indirectly inferred from a USGS water level gauge located along the coast. Similarly, the boundary conditions at the ocean in the Atchafalaya model was indirectly inferred from a NOAA station located off-shore. Although it was not used here for this purpose, AirSWOT can also be used to verify discharge boundary conditions in models where a riverine discharge is imposed (Zhang et al., 2022a).

The validation of the water levels across the domain using in-situ measurements further confirms the goodness of the calibrated friction coefficient using AirSWOT (Figures S4). Despite the lack of spatial information, timeseries can fill the temporal gap of remote sensing imagery. A combination of both types of data is a powerful tool for a holistic calibration and validation of numerical models.

## 4.2 UAVSAR

The difference between observed and modelled changes in water level over a fixed time was employed to correct marsh topography. If such difference is large, more flow conveyance is needed to increase water level changes, and vice versa. This can be done by modifying either the marsh elevation or the drag coefficient or both. As suggested by Zhang et al. (2022a), friction plays a marginal role in affecting water levels on marsh platforms. They run a sensitivity analysis with a wide range of friction values and found little effect on model performance. The calibration of topography inherently contains information on friction, which can lead to its small effect on the computed flow field. Applying the same iterative method to friction only, without modifying marsh elevation, would lead to unrealistically large spatial variations of the friction coefficient. Therefore, it was decided to only change marsh elevation to match modelled water-level variations with those derived via UAVSAR.

The ability of a model to correctly simulate water fluxes is tightly dependent on having accurate and precise marsh topography. This is especially true in coastal Louisiana where tidal ranges are small. Here, even a small topographic change can lead to notable variations in the flooding period (Donatelli et al., 2023a). Bathymetric data from LiDAR may present a positive bias in vegetated areas due to the inability of laser pulses to penetrate dense thick vegetation (Rosso et al., 2006). Furthermore, such bias is spatially variable and depends on vegetation characteristics (Medeiros et al., 2015). UAVSAR data can fill this gap. A simple comparison between model results and remote sensing data allowed precise identifications of critical parts of the domain and improvement of water fluxes on the marsh. The adjustment of the marsh elevation could be also carried out by adding or subtracting a constant value. However, UAVSAR allows to adjust the elevation at each point based on local hydrodynamics, unraveling the effect of micro-topography on water flow.

Starting from the initial topography, at each iteration the RMSE decreases until the minimum value of 3.9 cm is reached. The error in the subsequent iterations increases as the procedure begins to over-correct the topography and introduces errors that result in worse performance. It is important to highlight that this methodology is empirical and might have some theoretical issues. For instance, by modifying the bed elevation at some location of a marsh, water levels will change in surrounding areas. Thus, this iterative procedure might introduce errors in areas where the topography is correct. The methodology also depends on how well the model can solve the tidal channels. This effect can be noted for the three points before mentioned where the marsh was deepened. In this case, the points are located in proximity to a narrow channel with a 1.5-2 m cross section. The 10 m resolution of the model represents a limitation because features such as channels and levees that are smaller than 10 m cannot be captured by the mesh. In this example, the UAVSAR flight line captured the flow during falling tide, and in this phase, areas of the marsh close to the channels drained faster than internal ones. Since the model does not capture these channels, the method tries to compensate by lowering the marsh to increase water fluxes even if the elevation is correct.

Another drawback of this approach is that the bathymetric correction is based on water-level changes detected by UAVSAR during a very short period between 19:29 and 22:59 UTC time on 12 April 2021. Such changes are only representative of the temporal window in which the UAVSAR campaign took place. Although the solution was improved, the choice of a different acquisition might yield a different final marsh topography.

## 4.3 AVIRIS-NG

The use of AVIRIS-NG is possible after the calibration and validation of the hydrodynamic model, since sediment concentration depends on both the properties of sediment particles and hydrodynamic forcing. For the large-scale Terrebonne model, the calibrated critical shear stress and settling velocity are comparable with those of previous studies in Terrebonne Bay (e.g., Liu et al. (2018)). Along the flight line, the lower portion of the model in the coastal waters of the Gulf of Mexico and the Caillou Lake portion in the middle provide the best match with the imagery, despite the fact that the modelled concentration tends to

be more homogeneous. Overall, considering the relatively low RMSE compared to the full range of sediment concentration, it is concluded that the model can capture resuspension of bottom sediments. The model is also able to reproduce the sediment plume entering Caillou Lake, although overestimating the concentration.

     A calibration using only measurements in the field presents limitations. The optimal set of parameters is $w_s = 0.325$ mm/s, $\tau_{cr,e} = 0.1$ Pa (Figure S6h), which differs from the best set found using AVIRIS-NG ($w_s = 0.25$ mm/s, $\tau_{cr,e} = 0.1$ Pa). Inter-

400 estingly, the validation shows that, despite the model tends to underestimate concentrations in both cases, the calibration with AVIRIS-NG provides the lowest RMSE (Figure S8). AVIRIS-NG provides the possibility to compare the results with spatially distributed data, allowing a more complete evaluation of model performance. Point measurements could also not cover the full range of concentration and lead to calibrated parameters not representative for the entire area. Especially in coastal areas, cohesive sediment properties are highly affected by flocculation. Fine sediment aggregate to form flocs, for which both settling

velocity and bed shear strength are highly uncertain and difficult to predict.

     In the Terrebonne large scale model there are also discrepancies in the northern areas and within the channel meander in the south-eastern portion of the flight line, where the model tends to overestimate sediment concentrations. These errors might be related to bathymetric modification of the mesh. To allow a correct tidal propagation, channels were deepened and enlarged, which might have generated flow velocities different from real ones. Errors can also be attributable to model assumptions that

are inherent to the developed model. Although Delft3D has 3D capabilities, it was opted for a vertically averaged 2D model to avoid high computational costs and potential numerical instability of 3D grids when wetting and drying tidal flats and salt marshes are included. Moreover, 3D models do not provide a substantial improvement of predictability in water levels with respect to 2D models (Bates, 2022).

     It is fundamental to consider that AVIRIS-NG measures TSS concentration at the water surface. Since the models are

415 depth-averaged, a well-mixed column is implicitly assumed, which is a condition that might not always be true. Along rivers, sediment concentration has been typically observed to increase from the surface to the bed (Lamb et al., 2020). Usually, the coarser fraction occupies the lower layers in the water column, while finer fractions are resuspended to the surface. Hence, since AVIRIS-NG can only characterize the water surface, TSS maps might not be representative of total transport, which represent

a limitation for the calibration of sediment properties. The well-mixed conditions might not also be present in sheltered areas. In these locations, vertical heterogeneity is promoted by density gradients and low energy conditions during low wind speed conditions (Kjerfve and Magill, 1989). The lack of sediment heterogeneity could also be a reason behind for the concentration underestimation in the western side of the flight line in the large-scale Terrebonne model. Here, calibrated simulations tends to be 30 mg/L lower compared to AVIRIS-NG data. It is possible that in those areas only the finest fractions (not included in the current granulometry) are entrained in the water column. Another possible source of error is the absence of secondary flows and 3D flow structures that might arise at the boundary between channels and marsh platform (Proust and Nikora, 2020).

The AVIRIS-NG data suggest further evaluations. The comparison between the wave and no-wave cases (Figures 9 and 10) shows that the inclusion of the wave-induced resuspension is fundamental to better capture SSC in the deltaic areas. This result is consistent with previous research that highlighted the important role of waves in the development of river deltas and the redistribution of sediment along the coastal inner shelf (Shi et al., 1997; Walker and Hammack, 2000; Corbett et al., 2007; Carniello et al., 2014). As a final note, it is worth to mention that maps of TSS from AVIRIS-NG can also be leveraged independently from the numerical models. For instance, TSS maps in the Wax Lake Delta reveal distinctive patterns called streaklines (Kundu et al., 2015) (see Figure 2c), that can be used to derive deposition/erosion patterns and flow velocity without the support of numerical models (Salter et al., 2022; Donatelli et al., 2023b).

## 5 Conclusions

This study shows the potential of calibrating numerical models using remote sensing imagery instead of traditional sparse field data. Images from two SAR sensors and one spectrometer sensor flown during the NASA Delta-X airborne mission were utilized for the calibration. UAVSAR measured water level change on the marsh platform, while AirSWOT derived water elevations in bays, lakes, channels, and AVIRIS-NG measured sediment concentrations.

AirSWOT was used to calibrate the bottom friction coefficient in Terrebonne and Atchafalaya Basins, UAVSAR data to correct bathymetric errors and improve wetland flooding, and TSS maps derived from AVIRIS-NG allowed to calibrate sediment transport parameters. The use of spatially extended remote sensing imagery enabled to quickly evaluate areas where models performed better and provided calibrated parameters that are consistent with previous literature values. Remote sensing data yield spatial information that point observations cannot capture. In the case of highly uncertain parameters, a calibration based entirely on fixed location measurements can lead to wrong values, while imagery from remote sensing can provide more spatially coherent dataset. At the same time, some limitations and considerations need to be accounted for when coupling imagery with a numerical model.

Finally, the use of water-level time series at different locations for the validation informed the temporal performance of the models. Given the limited number of flyovers, temporal data allow to verify the accuracy of the calibration and test whether the model reproduces coherent water elevations over a wider time window in specific points of the domain.

The array of available remote sensing data will grow in the near future. This study showed that a new generation of numerical models can be developed by leveraging the spatial information provided by remote sensors.

*Data availability.* All Delta-X products and data used in this study are deposited at the correspondent ORNL-DAAC repository (https://daac.ornl.gov/daacdata/deltax/). Water level data from the Coastwide Reference Monitoring System (CRMS) used for the large scale Terrebonne model validation and small scale Terrebonne model boundary conditions are publicly available via the CPRA website (https://cims.coastal.louisiana.gov/monitoring-data/). Water levels data from the USGS station 073813498 used as boundary conditions in the large scale Terrebonne model are available via the USGS website (https://waterdata.usgs.gov/monitoring-location/073813498/#parameterCode=00065&period=P7D). Water discharge and sediment concentration data for the Atchafalaya model from the USGS stations 07381590 and 07381600 are available via the USGS website (https://waterdata.usgs.gov/monitoring-location/07381590/#parameterCode=00065&period=P7D) and (https://waterdata.usgs.gov/monitoring-location/07381600/#parameterCode=00065&period=P7D) respectively. Water level data from the Eugene Island NOAA station used as oceanic boundary condition in the Atchafalaya model are available via the NOAA website (https://tidesandcurrents.noaa.gov/stationhome.html?id=8764314).

*Author contributions.* LC, CD, XZ performed the simulations and interpreted the results. SF supervised the project. LC wrote the manuscript with contributions from all co-authors.

*Competing interests.* The authors declare that there is no conflict of interest regarding the publication of this article.

*Acknowledgements.* This research was funded by the NASA Delta-X project (the Science Mission Directorate's Earth Science Division through the Earth Venture Suborbital-3 Program NNH17ZDA001N-EVS3). LC was supported by the Future Investigators in NASA Earth and Space Science and Technology (FINNEST) award number 80NSSC21K1612. SF was also supported by the Virginia Coast Reserve Long-Term Ecological Research Program (National Science Foundation DEB-1832221) and the Plum Island Ecosystems Long-Term Ecological Research Program (National Science Foundation OCE-2224608). This work was carried out in part at the Jet Propulsion Laboratory, California Institute of Technology, under a contract with the National Aeronautics and Space Administration. We are grateful to Michael P. Lamb, Paola Passalacqua, Robert Twilley, and all Delta-X team members that contributed to the conceptualization of the Delta-X mission and the data that included in this study.

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
