# Peer review of "Coupling numerical models of deltaic wetlands with AirSWOT, UAVSAR, and AVIRIS-NG remote sensing data"

_Biogeosciences, 2023_

## Author Comment (AC1)

**Comment on bg-2023-108**

**Reply to RC1**

The paper discusses the use of 3 different remote sensing products for the calibration and validation of 3 hydrodynamic models of different scales: AirSWOT to calibrate friction coefficient, UAVSAR to calibrate errors in the intertidal marsh topography and AVIRIS-NG to calibrate sediment properties.

The authors introduce the traditional calibration techniques for such a model, being the use of time series of observations (e.g. water levels, sediment concentration, etc.) and emphasise how remote sensing observations can complement such technique. I believe they innovative character of the research is clearly presented.

We want to thank the Reviewer for providing feedbacks and comments, which greatly helped better presenting our findings. In the following lines, we reply (in **black**) to each comment (in **blue**) and refer to the changes in the manuscript. Modifications are reported between "" and with *italicized* text. At the end, we also report all references to papers cited in the answers. We were not able to add all revisions. In these cases, we refer to the changes we will make in the manuscript instead of providing the actual modification.

**Major comments:**

Regarding the use of the AirSWOT data: both the calibration and validation are clearly presented. However, whether the use of AirSWOT actually leads to more accurate model results than calibration using time series is less obvious. Can you calibrate the model using the airswot and using time-series to show in a validation period that the airswot actually appears to be better? In the discussion, could you compare your evaluation statistics with other similar hydrodynamic models of intertidal areas? What are typical RMSE values? If needed, the authors could also calculate other evaluation statistics such as the Nash & Sutcliffe model efficiency to allow comparison with more existing studies.

We extensively updated this part of the paper. We followed the suggestion and performed two separated calibrations using AirSWOT and timeseries. By doing these we also realized that we could achieve better performance using a friction coefficient of 65 $m^{1/2}s^{-1}$. In the new version of the manuscript, we use the same ranges evaluated in the Atchafalaya model and compare results between 45 and 65 $m^{1/2}s^{-1}$. By doing so, we were also able to detect a few small calculation errors we previously made. Interestingly, we see that in the calibration with AirSWOT, 65 $m^{1/2}s^{-1}$ is the best value, while in the timeseries calibration we have similar results but slightly better in the 45 $m^{1/2}s^{-1}$ case. We then use the two coefficients in the Delta-X Fall campaign to validate the water levels and find that 65 $m^{1/2}s^{-1}$ works better by comparing water levels gauges. We used the RMSE and Model Efficiency index as suggested to compare performance. In the discussion, we added a few comparisons with existing studies to confirm the goodness of our results.

UAVSAR: The method is only very briefly explained how the use of water surface elevation changes can be used to calibrate errors in the marsh topography and the authors refer to another paper where more details can be found. As this is an essential part of the paper, I would like to suggest a slightly more extensive explanation of this method. Furthermore, could you address how the UAVSAR - topography calibration can be validated and why no direct validation of the calibrated parameter (being the marsh topography) is included in the paper? Finally, assuming a uniform friction coefficient is very likely not to be the case in reality and how do you ensure that

the calibration of the marsh topography does not try to resolve these types of errors instead of errors in the marsh topography.

We agree on the fact that the procedure is important and with the suggestion of providing directly more details instead of referring to the paper that developed the method. To address this, instead of presenting the principle behind the method, we now explicitly explain the iterative procedure, so that it is more understandable to the reader. Furthermore, we added a new figure in the Supplementary Material, in which we provide a simple flow diagram showing the method (the figure is adapted from Zhang et al., (2022a)). The new method description reads: "*To correct marsh topography, the methodology proposed by Zhang et al. (2022a) was followed, in which errors in topography were removed by comparing modelled water-level changes with those observed via UAVSAR (see Figure S1). In this procedure, the first simulation is run using the original topography and the difference between modelled and observed water level change is computed in each cell. If the modelled water level change is larger than UAVSAR, the elevation of the cell is increased. In the opposite case, the elevation is decreased. The updated topography is used in the subsequent simulation. The procedure is run with the updated topography iteratively until the minimum RMSE is reached*".

One way to validate the modified bathymetry could be comparing the water level change in a different time using a different UAVSAR flight line. More specifically, one could consider a water level change measured between two different times. In our case, we chose a change in 3.5 hours. In the description of the procedure, Zhang et al. (2022a) point out that as long as the method converges to a unique solution with realistic values, using a different time would lead to similar results. Another way to validate is considering ground-truth values of elevation measured in-situ. In previous version of the manuscript we did not include any validation, however, in the new version we provide a validation of the results using RTK measurements collected during the Delta-X Spring 2021 campaign and RTK measurements collected by the CRMS. Despite some points show that the original topography was closer to the real value, the coupling with UAVSAR allowed us to reduce the error. It must be noted that the measurements cover a very limited area of the domain. A better validation would consider more spatially distributed point; however these are not available. We added a sentence in the subsection 2.5: "*To validate the topographic correction we used compared the original and calibrated topography with Real Time Kinematics elevation measurements collected during the Delta-X Spring 2021 campaign (Twilley and Rovai, 2022) and site 421 elevation provided by the CRMS*". We decided to show the validation figure in the Supplementary Material (Figure S2).

Regarding the friction we added a more detailed point in the discussion in the manuscript. Drag exerted by bottom and vegetation effects the flow. Zhang et al. (2020) showed that model performance did not significantly change for different Chezy coefficients, showing that elevation in these areas exerts a strong influence. They point out, as the Reviewer suggests, that the correction of the topography contains information of the friction. Moreover, they observed that if one would run the calibration process only on friction, this would lead to unrealistic values of frictions and unrealistic spatial distribution. Thus, a good practise would be to set an initial friction, calibrate the topography, and then adjust for friction by paying attention to stay reasonable ranges. In this case, we did not calibrated friction. We added a few sentences in the discussion of UAVSAR: "*As suggested by Zhang et al. (2022), friction plays a marginal role in affecting water levels on marsh platforms. They run a sensitivity analysis on the effect of variable friction by exploring a wide range of values finding little effect on model performance. The calibration of the topography inherently contains information of the friction, which can lead to its smaller effect on the computed flow. At the same time, applying the same procedure to the friction (without modifying the marsh elevation) would lead to unrealistic large spatial variation of the friction coefficient, with some unrealistic*

*values. Therefore, it was decided to only change marsh elevation to match modelled water-level variations with those derived via UAVSAR".*

AVIRIS-NG: While the calibration process is clearly explained, I believe more emphasis should be put on validating whether the use of remote sensing indeed improves the model over the use of single point data. The author mentions in-situ observations. I would suggest calibrating the model using the in-situ measurements and independently calibrate the model with a AVIRIS-NG image. Then, independent validation (based on either in-situ measurements and/or AVIRIS-NG imagery of a different time period) could indicate whether the model calibrated with the AVIRIS-NG image indeed performs better.

We agree with the evaluation of the reviewer. More effort should be put in the validation of the suspended sediment. As suggested, we will use the TSS measurements that cover our area. We will add an independent calibration with in-situ measurements and validate the results of both calibration using measurements in the Delta-X Fall campaign. We will add the new figures in the Supplementary material.

**Minor comments:**

1. Line 134: Could you add the uncertainty on water level measurements.

   For each flight line there is an error file (named *err* in the Denbina et al. (2022) dataset) which contains an estimate of the vertical error for each pixel and provides a spatially-varying estimates based on the interferometric correlation. For both Delta-X campaigns, AirSWOT was validated using in-situ gauges and it was found a Root Mean Squared Error of 9 cm across the entire campaigns when AirSWOT was averaged over a 1 km$^2$ area. We added the last information in the manuscript at the end of the AirSWOT data presentation: "*Water surface elevation data were validated in both Delta-X campaigns using in-situ gauges and a root mean squared error of 9 cm was found when data were averaged on a 1 km$^2$ area*".

2. Line 108: Could you explain why small-scale Terrebonne model domain was chosen as a small-scale region of interest? Could you support that this is a representative area?

   Since the large-scale Terrebonne model covers a very large area (about 90 km east to west), it has a coarse resolution in order to be more computationally efficient. The downside is the inability to reproduce small features. Given the microtidal range, differences in water level change on the marsh depends on small topographic differences, that such a coarse grid cannot capture. Hence, the small-scale model allowed us to incorporate UAVSAR data in our numerical modelling framework. As we mentioned at the beginning of paragraph 2.1, the Terrebonne Basin has been constantly losing marshland. The small-scale domain is located in these degrading salt marshes. Thus, the processes in these areas can be considered representative for most of the Terrebonne Basin.

3. Line 164: could you support the decision for the chosen Chezy coefficients for ocean and marsh platform and on line 186 could you explain why these values differ from the large-scale Terrebone model? Same comment for the values mentioned in line 206.

   The Chezy coefficient range is chosen based on different modelling studies of coastal marshes and deltas. We forgot to add those references. We added a new sentence indicating this: "*The selected values fall within a range considered by several coastal marshes and deltas modelling studies (e.g. Edmonds and Slingerland (2010); Nardin eta al. (2013); Stark et al. (2015); Zhang et al. (2019))*". This applies for both Terrebonne and Atchafalya.

In the new version of the manuscript we improved the AirSWOT analysis and now we explore the same friction range in the two large-scale models. A different Chezy coefficient in the channels in the small-scale model in Terrebonne was used simply because the two models were developed at different time. However, the differences between using 55 or 65 $m^{1/2}s^{-1}$ are minimal and do not significantly affect model results.

1. Figure 4 & Figure 7: Spatial patterns are difficult to observe, could you make the map larger (for instance, by rotating the map so that the flight line is either horizontal or vertical)? Could you change the colour scale to stretch the values in the raster map? It seems both negative and positive values on the difference maps never - and + 0.75 m.

   We have modified both figures to better observe the patterns. We have enlarged the flight line in Figure 4 to focus on the south area of the flight line (similarly to Figure 5). Now the figure is clearer. We have also decided that in order to be consistent with the Figure 5, we show only one flight line instead of four (this allowed us a clearer figure). We included a figure with the four flight lines in the supplementary material so the reader can see a comparison with the same flight line in different time of the tidal cycle. We have also reduced the ranges of the error colorbars.

2. Figure 5 and 6: Could you make these maps also bigger? I would propose to only show the colorbar for elevation once and the error colorbar once and if needed, drop the grid labels and add a scale (like figure 4). The scale and zoom of figure 2 is a good example. The use of a single colorbar is shown very well in figure 7. In figure 5, subfigures are not labelled.

   We have modified Figure 5 and Figure 6 and followed the suggestions. Both figures are now larger and more readable. There is only one colorbar for the evelation and error in Figure 5 and only one colorbar for the water level change in Figure 6. In the latter, we also added a land/water mask to better separate channels and marsh and change the order of the subfigures with a better one. We added labels where missing.

3. Line 257: unclear which areas are considered critical and why.

   The critical areas are mentioned in the previous lines: the bottom an upper area. However, it was difficult to connect them to the 'critical areas' later mentioned. Those are considered as such because they show the higher disagreement between simulation and UAVSAR after the first run. We modified the text to correct the unclear sentences. Now the paragraph reads: "*Results from the first run (Figure 6e and 6g) highlight two critical areas. First, the model overestimates the water level change in the bottom area, indicating an error in the marsh elevation derived from LiDAR data. The opposite is verified in the northern area, where waters are found to recede too slowly and consequently underestimate the water level change*".

4. Line 309: could you refer to the specific figure which supports this statement.

   We forgot to refer to the Figures in the Supplementary Information. In this case the correct reference would have been Figures S1 and S2. However, in the new version of the manuscript we removed these figures to show the results of the calibration with timeseries and validation of the calibrated Chezy in a different period. We will modify the text accordingly with the new references.

5. Line 267: switch order or 'always' and 'to'.

   Corrected. The sentence now reads: "Here, the model tends to always overestimate sediment concentration and the error decreases as critical shear stress and settling velocity increases".

6. Line 268: switch order of 'better' and 'performs'.

Corrected. The sentence now reads: "Overall, the model performs better in the more open sections of the flight line, while at the extremities, performance declines".

7. Line 269: drop the first 'best' and the sentence misses an active verb (online 2017 change 'as' to 'is').

Corrected. The sentence now reads: "The combination that provides the best comparison with measurements is a critical shear stress of 0.1 Pa and a sediment settling velocity of 0.25 mm/s".

**References**

Denbina, M., Simard, M., and Rodriguez, E.: Delta-X: AirSWOT L2 Geocoded Water Surface Elevation, MRD, Louisiana, 2021, Version 2. ORNL DAAC, Oak Ridge, Tennessee, USA, https://doi.org/10.3334/ORNLDAAC/2128, 2022.

Zhang, X., Jones, C. E., Oliver-Cabrera, T., Simard, M., and Fagherazzi, S.: Using rapid repeat SAR interferometry to improve hydrodynamic models of flood propagation in coastal wetlands, Advances in Water Resources, 159, 104088, https://doi.org/10.1016/j.advwatres.2021.104088, 2022a.

---

## Author Comment (AC2)

**Comment on bg-2023-108**

**Reply RC2**

This study tries to improve numerical model predictions and validation of sediment fluxes between tidal marshes, channels and bays, by data assimilation of high resolution remote sensing imagery.

This topic is worthwhile and important to explore, timely and well suited for the audience of biogeosciences.

The study is of high quality and well written.

We want to thank the Reviewer for taking the time to carefully read our manuscript and providing comments to better present our work. In the following lines, we reply (in **black**) to each comment (in **blue**) and refer to the changes in the manuscript. Modifications are reported between "" and with *italicized* text. At the end, we also report all references to papers cited in the answers. We were not able to add all revisions. In these cases, we refer to the changes we will make in the manuscript instead of providing the actual modification.

Line 65: could more information been added here. I assume the authors are referring to surface concentrations of suspended solids derived from AVIRIS-NG

Yes, we refer to the **surface** sediment concentration. We modified the sentence to include this information. In addition, we added an information on the method used to derive those data: "*Jensen et al. (2019) used high resolution remote-sensing reflectance data from NASA's Airborne Visible/Infrared Imaging Spectrometer-Next Generation (AVIRIS-NG). They developed an algorithm based on a derivative-based partial least squares regression between measured total surface suspended solids and in-situ spectra, and} derived maps of total surface suspended solids in the waters of the Atchafalaya basin along the Louisiana coast (USA) from AVIRIS-NG*".

Line 95: this might be just a detail but I suggest to put species names in italic

Using italicized text is more precise, so we applied the suggested modification.

Line 125ff: how can the UAVSAR and AirSWOT distinguish between emerged vegetation or the water surface? Are there vertical errors ranges that could be reported?

In UAVSAR, in order to separate wetlands and water surface, a water mask is generated from the interferogram L1 products. AirSWOT mounts a digital camera that maps the water surface. However, in the case of the Delta-X mission the camera wasn't mounted as it requires cloud-free conditions. Since UAVSAR is better at discriminating water and wetlands, the UAVSAR water mask was used also for AirSWOT. We added two sentences specifying this: "*In flooded wetlands, the water surface is detected through the double-bounce scattering mechanism from water and vegetation (Kim et al., 2009; Wdowinski et al., 2013). To separate the water surface from the emergent wetland, a water mask was generated from the interferogram Level-1 products*". In the AirSWOT data introduction we added: "*AirSWOT uses cross-track interferometry to measure the elevation and combines it with along-track interferometry to correct for the bias due to the water motion (Goldstein and Zebker, 1987). To separate land from water surface, the same UAVSAR water mask was used for AirSWOT*".

Regarding the vertical error, there is an error file (named *err* in the Denbina et al. (2022) dataset) which contains an estimate of the vertical error for each pixel and provides a spatially-varying

estimates based on the interferometric correlation. For both Delta-X campaigns, AirSWOT was validated using in-situ gauges and found a Root Mean Squared Error of 9 cm across the entire campaigns when AirSWOT was averaged over a 1 km$^2$ area. We added the last information in the manuscript at the end of the AirSWOT data presentation: "*Water surface elevation data were validated in both Delta-X campaigns using in-situ gauges and a root mean squared error of 9 cm was found when data were averaged on a 1 km$^2$ area*".

Line 144ff: is TSS surface or depth averaged concentration derived? How applicable is the calibration during different types of suspended particles during high and low flow conditions?

The TSS derived from AVIRIS-NG imagery is surface concentration. The in-situ measurements used to develop and validate the algorithms were collected in a wide range of contrasting water types in Terrebonne and Atchafalaya Bay during both Delta-X campaign, which covered high flow (during Spring 2021) and low flow (during Fall 2021) conditions. The algorithm to retrieve TSS from AVIRIS-NG performed well (Median Absolute Percent Difference 13.7% and Median bias 6.71 mg L$^{-1}$) across a wide range of TSS concentrations (0.1–154.5 mg L$^{-1}$) throughout both basins and in contrasting seasons. More detailed information on the spatial and temporal coverage of the sampling and algorithm calibration are in Fichot and Herringmeyer (2022) and Fichot and Herringmeyer (2023). We modified the paragraph related to AVIRIS-NG data and included these informations: "*Local empirical algorithms derived using in-situ measurements were used to derive TSS concentration form the Rrs(λ) in the visible/near-infrared region and generate maps of TSS from the AVIRIS-NG imagery (Gao et al., 1993; Bue et al., 2015; Jensen et al., 2019). The in-situ samples were collected in both Terrebonne and Atchafalaya basins during the Delta-X 2021 Spring and Fall campaigns in order to capture high and low flow conditions. The algorithm to retrieve TSS from AVIRIS-NG performed well (Median Absolute Percent Difference 13.7\% and Median bias 6.71 mg/l) across a wide range of TSS concentrations (0.1–154.5 mg/l) (Fichot and Herringmeyer, 2021, 2022)*".

Line 159: what was the grain size, what transport equations were used?

We set the sand median diameter at 0.14 mm. For the cohesive fraction, we defined the settling velocity and not the grain size. The sand transport is modelled using Van Rijn (2007), while the mud transport is modelled with Partheniades-Krone (Partheniades, 1965). A few of this information was already included, however a part was not indicated. Now we explicitly state the equation used. "*Thus, the calibration of the parameters refers to the properties of the mud fraction. The default transport equation of Van Rijn (2007) was used for sand*". "*The default Delft3D sediment transport formulation of Partheniades-Krone (Partheniades, 1965) was used for the non-cohesive fraction*".

How was the bed initialized, 1- or multiple layers?

The bed is initiated as 1 layer. In each cell we defined a percentage of sand and mud based on Williams et al. (2006).

What was the active layer thickness?

The layer thickness was set at 5 m to ensure we did not run out of sediment to resuspend.

Was the option for mixed sediments used?

We did not use the option for mixed sediments and two fractions were modelled separately.

How was the non-cohesive/cohesive boundary defined?

The separation between sand and mud (non-cohesive and cohesive fraction) was based on a previous study in the area (Liu et al., 2018), in which they model deposition generated by Hurricane Gustav. Median sand diameter size is set to 0.14 mm, which locates the sediment in the fine sand class (using the Wentworth scale). Everything below is considered cohesive.

Line 165: is it correct to assume that for tidal channels/lakes and bays the same chezy coefficient was used? did the author try a spatial varying chezy? which wetting and drying scheme was used?

This is a very interesting point that was discussed during the development of the model. The assumption of a homogeneous Chezy coefficient holds some simplifications and initially we tried to use a spatially varying Chezy coefficient to improve the comparison with AirSWOT, especially for the smaller areas and channels within the marsh. However, we realized that for the large-scale models, results did not improve by varying the value of the friction. The limitations of the coarse grid outcompeted those due to the same friction coefficient. Namely, before considering more tuning of the friction, the model must correctly solve the flow in smaller areas and channels. In a possible future improvement of the models, using an unstructured grid could allow better representing smaller features. For instance, a separation in terms of friction could be done between ocean/main tidal channels and small tidal creeks.

The wetting and dry scheme is based on a threshold depth which was set at 1 cm. The algorithm 'activates' a cell when the water depth is positive and larger than the threshold, while the cell becomes dry wet the water depth falls below the threshold. More precisely, the algorithm uses half of the wetting threshold to avoid a "flip-flop" (a change of state in two consecutive time steps, due to oscillation generated by the algorithm). We added a sentence in subsection 2.5: "*A 1 cm was set as water depth threshold for the wetting-drying scheme*". We added the sentence here since we focus on marsh flooding only in the small-scale model.

Line 170: I assume ws is calculated from the median diameter and the sediment density I assume the parteniades krone relation is used for sediment pickup please indicate? See comment above what equation was used for sand.

Yes, settling velocity of sand is computed as function of the median diameter and density according to Van Rijn (1993), where one out of three different equations is used depending on the median diameter. We added the information in the manuscript: "*Note that in the case of non-cohesive particles, Delft3D does not require to specify a value for the settling velocity, since it is directly computed from the median diameter and density using the Van Rijn (1993) approach*".
The Partheniades-Krone formulation was used for the cohesive fraction. Please refer to the previous answer for more details and related modification to the manuscript.

Line 175: similar comment as above why was only one muddy sediment class in the bed considered? For instance consolidated clay lenses can possess high crit. bss.,, .. did something like that occur? how well was the initial stratigraphy incorporated in the model?

We acknowledge that within our mud class multiple subclasses are present, thus our separation represents a simplification of the real conditions. The main reasons we opted for a two-class separation, is that the only robust dataset we have to represent the bottom (Williams et al., 2006) provides sampling points were most of the time this is the only separation. As pointed out in an earlier comment, the initial stratigraphy is based on this dataset.

Line 185: is the marsh platform chezy also representing vegetation?

In this case the Chezy coefficient does not account for vegetation but only the bottom of the marsh. The 35 $m^{1/2}s^{-1}$ is a typical value used to represent bottom roughness in modelling studies (e.g. Zhang et al., 2018; Passeri et al., 2018). In order to include vegetation a more correct value would range between 10 and 20 $m^{1/2}s^{-1}$ (e.g. Augustijn et al., 2008; Stark et al., 2015) or implement formulations such as Baptist (2005). In this case, our objective was to calibrate elevation and not friction. Calibrating only friction instead of wetland elevation would have led to unrealistic spatial distribution and values of Chezy (Zhang et al., 2022). Indeed, the authors suggest to first calibrate boundary conditions and elevation. Only after these steps, a calibration of the friction would provide more realistic values of friction. For this reason, we simply set an initial and homogeneous friction coefficient. Note that, the calibration of the elevation inherently contains information of vegetation, however, when Zhang et al. (2022) ran a sensitivity analysis on marsh Chezy coefficient found non-significant variation of model performance for all Chezy values (range 8-40 $m^{1/2}s^{-1}$).

Line 199: what kind of data was provided.. suspended sediment concentration,..?

Yes, it is suspended sediment concentration data. We were not precise, so we add the 'concentration' term in the sentence.

Line 205: was the initial sediment distribution uniform...? was a spinup for the bed tested?

Yes, the Atchafalaya model initial sediment distribution on the bed was taken uniform and it is based on the usSEABED database. In this case we have fewer points compared to Terrebonne. A spin-up of the bed is run for about one year with a morphological speeding factor of 50 to reach the equilibrium state. During this process, the bed level is kept fixed, and the bed fraction changes to adapt to the hydrodynamics. The bed composition after the spin-up process becomes more realistic and coarser fractions appear in areas with stronger flow shear stress. We followed an approach consistent with previous studies such as van der Wegen et al. (2011) and Zhang et al. (2020). We have added this information in the subsection 2.6: "*For sediment transport, three sediment types composing the bottom: sand, silt, and mud were considered. The initial sediment distribution was derived from the usSEABED database and set uniform at 22\% sand, 39\% silt, and 39\% clay. A bed spin-up process of one year was run with a morphological speeding factor of 50 to reach the equilibrium state. During this process, the bed level was kept fixed, and the bed fraction changed to adapt to the hydrodynamics. The bed composition after this spin-up process became more realistic and the coarser fractions appeared in areas with stronger flow shear stress (see similar approaches in van der Wegen et al. (2011) and Zhang et al. (2020))*".

Line 215: see comment above why was no spatial tunning test? this could also improve fig.5?

Regarding the spatial tuning of the friction for Figure 5, first, it is important to mention that the area in dark blue colour on the left side of the flight line is affected by the edge effect. The primary instrument of AirSWOT is the interferometric synthetic aperture radar called KaSPAR. The KaSPAR outer swath mode has incident angle between 4 and 25°. This means that at the edges of this range we find the largest error (Denbina et al., 2019). At the edges the water surface elevation might not be correct. This is something we did not mention in the discussion. We will add a sentence to provide the information to the reader. Another point regarding the friction, is that the friction is typically defined for broad classes, in particular for areas that are constantly underwater such as channels. A reasonable change in friction coefficient (meaning changing the values to physically sound values) would not particularly change the water levels.

Line 240: is the RMSE calculated over an entire M2 tide or only during the time-slice the picture was taken?

The simulations were run over the entire Delta-X Spring 2021 campaign in order to check for temporal performance of the models. For simplicity, in the paper we mention only the Terrebonne model. The RMSE we refer here is computed considering all measured and modelled water levels in 13 water level gauge of site of the CRMS network (see Table S1). This value is not related to the RMSE used with the spatial images. The last one quantifies the goodness for every single comparison.

Fig.7: although the model result are very impressive, the predicted error in SSC is still between 30% - 50% of the measured range, for what conditions in the tide is this representative, is this the best or worst case?

This flight line was collected on 05 April 2021 at 19:57 UTC. This is approximately one hour after low tide when the tide is rising. The choice of the flight line was mostly dictated by the quality of the retrieved TSS. Cloud conditions were optimal (high quality flight line) and in the area covered by the flight line, there is one of the sampling points used to calibrate the algorithm used to retrieve TSS from AVIRIS-NG. Thus, this is one of the best cases we could show of all the available flight lines.

I think fig.7 could be improved since open water or marsh area are difficult to distinguish? If I interpret the results correctly the biggest error seem to appear in shallow areas?

We have improved Figure 7 by making it larger and more readable. The second question could be another interesting point to add to the results. To give a more precise answer we will try to plot the error as function of the depth and see if it increases as depth decreases.

Line 295: was a finer mesh tested,.. was an unstructured grid tested?

In very preliminary test we tested a 50 m grid, which did not provide significant improvement but only enhanced computational costs. We did not implement an unstructured grid. This is a limitation of the model, which can reproduce water levels in the wider open areas but struggles to resolve smaller channels. This limitation can be partly addressed by switching to an unstructured grid. It has also to be noted that the unstructured grid would have limitations. Many small channels and creeks are about 1-2 m wide. Such a small size would be challenging also for an unstructured grid.

Line 305: see comment above,.. very low water levels and gradients,.. will make the mini. water depth and flooding and drying scheme relatively important?

The reviewer is correct. As we mentioned in a previous answer the wetting-drying scheme is based on a threshold depth, thus the minimal water depth is very important. Delft3D-FLOW uses an element removal algorithm. For instance, the FVCOM model, which is employed by the Deb et al. (2023) paper introduced in the next question, uses a thin film algorithm (Medeiros and Hagen, 2013). Using a different wetting and drying method would affect the final result, thus it is important to provide the information in the methods section. It is worth noting that, this observation is only valid for the marsh area that are undergo flooding and drying, therefore only the application of UAVSAR. The other applications are related to channels and bays which are always underwater (no drying involved).

Line 320: how did the marsh microtopography influence the wetting and especially drying of the interior, i.e. Deb 2023,.. used an additional porosity to limit ponding caused by submesh channels?

As we mention in the manuscript, due to the very small tidal range, the topography of the marsh plays a crucial role in the wetting and drying. The suggested paper introduces to a clever way to deal with submesh channels, which is a problem also in our case. The authors modified FVCOM, which is very different from the structured grid version of Delft3D we used. As we said a few answers earlier, this is a limit of the models we developed. The problem of artificial ponding caused by submesh channels that are not solved is indeed a limitation of the methods used to correct the bathymetry. The method depends on the ability of the model to solve the water fluxes. Therefore, if some areas are not well solved, it could be that the topography is changed even if it is not necessary. For instance, let's suppose there is a small channel that the 10 m mesh cannot represent. During falling tides, the areas near the channel drain faster, hence, they generate a larger water level change compared to internal areas. Since the model cannot solve the channel, the procedure will deepen the topography to allow water to flow and match the water level change, even if the topography is correct. We added this discussion by leveraging on some RTK measurements that we used to validate the calibrated topography: "*this iterative procedure might introduce errors in areas where the topography is correct. This effect can be noted for the three points before mentioned where the marsh is deepened. The methodology depends on how well the model is able to solve all channels. In this particular case, the points are located in proximity to a narrow channel with a 1.5-2 m cross section. The 10 m resolution represents a limitation because features such as channels and levees that are smaller than 10 m cannot be captured by the mesh. Yet, they affect water flow. In this example, we used a UAVSAR captured during falling tide, and in this phase, areas of the marsh close to these channels drains water faster than areas located internally. Since the model does not capture these channels, the method tries to compensate by lowering the marsh to achieve the measured water level change even if the elevation is correct*".

Line 370: it is unclear how the davg sediment concentration predicted by D3d was correct to compare to the remote sensing product.

This is an important point that was also raised by the other reviewers, and we agree that the current version of the paper does not clearly show it. We will perform a second calibration using in-situ measurements and provide a validation of the results using a second period (Delta-X Fall campaign).

**References**

Augustijn, D. C., Huthoff, F., & Van Velzen, E. H. (2008). Comparison of vegetation roughness descriptions. *Altinakar, MS, Kokpinar, MA, Aydin, I., Cokgor, S. Kirkgoz, S.(eds.) River Flow*, *2008*, 343-350.

Baptist, M.J., 2005. Modelling Floodplain Biogeomorphology. Delft University of Technology, TU Delft.

Denbina, M., Simard, M., and Rodriguez, E.: Delta-X: AirSWOT L2 Geocoded Water Surface Elevation, MRD, Louisiana, 2021, Version 2. ORNL DAAC, Oak Ridge, Tennessee, USA, https://doi.org/10.3334/ORNLDAAC/2128, 2022.

Denbina, M., Simard, M., Rodriguez, E., Wu, X., Chen, A., and Pavelsky, T.: Mapping water surface elevation and slope in the mississippi river delta using the AirSWOT Ka-Band interferometric synthetic aperture radar, Remote Sensing, 11, 2739, https://doi.org/10.3390/rs11232739, 2019.

Fichot, C.G., and J. Harringmeyer. 2022. Delta-X: AVIRIS-NG L3-derived Water Quality, TSS, and Turbidity, MRD, LA 2021, V2. ORNL DAAC, Oak Ridge, Tennessee, USA. https://doi.org/10.3334/ORNLDAAC/2112

Fichot, C.G., and J. Harringmeyer. 2022. Delta-X: In Situ Water Surface Reflectance across MRD, LA, USA, 2021, Version 2. ORNL DAAC, Oak Ridge, Tennessee, USA. https://doi.org/10.3334/ORNLDAAC/2076

Liu, K., Chen, Q., Hu, K., Xu, K., and Twilley, R. R.: Modeling hurricane-induced wetland-bay and bay-shelf sediment fluxes, Coastal Engineering, 135, 77–90, https://doi.org/10.1016/j.coastaleng.2017.12.014, 2018.

Medeiros, S. C., & Hagen, S. C. (2013). Review of wetting and drying algorithms for numerical tidal flow models. *International journal for numerical methods in fluids*, *71*(4), 473-487. https://doi.org/10.1002/fld.3668

Partheniades, E.: Erosion and deposition of cohesive soils, Journal of the Hydraulics Division, 91, 105–139, https://doi.org/10.1061/JYCEAJ.0001165, 1965.

Passeri, D. L., Long, J. W., Plant, N. G., Bilskie, M. V., & Hagen, S. C. (2018). The influence of bed friction variability due to land cover on storm-driven barrier island morphodynamics. *Coastal Engineering*, *132*, 82-94.

Stark, J., Van Oyen, T., Meire, P., & Temmerman, S. (2015). Observations of tidal and storm surge attenuation in a large tidal marsh. *Limnology and Oceanography*, *60*(4), 1371-1381.

Van der Wegen, M., Dastgheib, A., Jaffe, B. E., & Roelvink, D. (2011). Bed composition generation for morphodynamic modeling: Case study of San Pablo Bay in California, USA. Ocean Dynamics, 61(2–3), 173–186. https://doi.org/10.1007/s10236-010-0314-2

Van Rijn, L. C. (2007). Unified view of sediment transport by currents and waves. I: Initiation of motion, bed roughness, and bed-load transport. *Journal of Hydraulic engineering*, *133*(6), 649-667.

Van Rijn, L. C. (1993). Principles of sediment transport in rivers. *Estuaries and Coastal Seas. Aqua Publications*.

Williams, S. J., Arsenault, M. A., Buczkowski, B. J., Reid, J. A., Flocks, J., Kulp, M. A., Penland, S., and Jenkins, C. J.: Surficial sediment character of the Louisiana offshore Continental Shelf region: a GIS Compilation, Tech. rep., US Geological Survey, https://doi.org/10.3133/ofr20061195, 2006.

Zhang, X., Leonardi, N., Donatelli, C., & Fagherazzi, S. (2019). Fate of cohesive sediments in a marsh-dominated estuary. *Advances in water resources*, *125*, 32-40.

Zhang, X., Leonardi, N., Donatelli, C., & Fagherazzi, S. (2020). Divergence of sediment fluxes triggered by sea-level rise will reshape coastal bays. Geophysical Research Letters, 47(13), e2020GL087862

Zhang, X., Jones, C. E., Oliver-Cabrera, T., Simard, M., and Fagherazzi, S.: Using rapid repeat SAR interferometry to improve hydrodynamic models of flood propagation in coastal wetlands, Advances in Water Resources, 159, 104088, https://doi.org/10.1016/j.advwatres.2021.104088, 2022a.

---

## Author Comment (AC3)

**Comment on bg-2023-108**

**Reply RC3**

I strongly agree with the central argument of this paper, that spatial information from remote sensing provides critical data for calibrating and testing numerical models of coastal wetlands. The paper reports on results of an experiment in which 3 remote sensing products were used in conjunction with modeling of a region of coastal Louisiana to evaluate the value of combining these approaches. Overall I think the analysis is thorough and informative, but the text would benefit from editing for clarity. Most of my comments are editorial.

We want to thank the Reviewer for positive support to our study. We are grateful for all comments, and editorial suggestions, which allowed us to better show our analysis. In the following lines, we reply (in **black**) to each comment (in blue) and refer to the changes in the manuscript. We report the related modification in the manuscript between "" and with *italicized* text. At the end, we also report all references to papers cited in the answers. We were not able to add all revisions. In these cases, we refer to the changes we will make in the manuscript instead of providing the actual modification.

Comments:

L18: "Peter Sheng et al." should just be "Sheng et al."

We doubled checked and the correct citation is Peter Sheng (see https://www.nature.com/articles/s41598-022-06850-z#citeas)

L19: There is a recent paper by Temmerman et al (2023; Reviews of Marine Science) that would be appropriate to cite here.

The suggested paper is very appropriate here, thus we added the citation.

L48: Does Gross Primary Productivity need to be capitalized?

No it does not. We corrected the mistake.

L60: change to: ", and coastal wetland above-ground biomass …"

Changed.

L106-112: The statement is made here that there are 3 models at different scales, but it seems that the scale of the large-scale Terrebonne model and the Atchafalaya model are essentially the same.

That is corrected. The large-scale Terrebonne model and the Atchafalaya model are developed at basin scale, and they are comparable (they also have the same spatial resolution). The paragraph was not precise, hence we added an additional sentence to clarify: "*Two models were set-up at the basin scale, while the third one was developed at smaller scale*".

L144: "TSS concentration from the …"

Corrected.

L145: Should the phrase "and vegetation structure are produced" be deleted? It seems the same point is made in the next sentence.

Yes. These lines were badly constructed. We reformulated and added more information regarding AVIRIS-NG and the algorithm to retrieve TSS: "*Local empirical algorithms derived using in-situ measurements were used to derive TSS concentration from the Rrs(λ) in the visible/near-infrared region and generate maps of TSS from the AVIRIS-NG imagery (Gao et al., 1993; Bue et al., 2015; Jensen et al., 2019). The in-situ samples were collected in both Terrebonne and Atchafalaya basins during the Delta-X 2021 Spring and Fall campaigns in order to capture high and low flow conditions. The algorithm to retrieve TSS from AVIRIS-NG performed well (Median Absolute Percent Difference 13.7\% and Median bias 6.71 mg/l) across a wide range of TSS concentrations (0.1–154.5 mg/l) (Fichot and Harringmeyer, 2021, 2022). AVIRIS-NG images were also used to produce maps of vegetation structure (Jensen et al., 2021)*".

L162: What is the resolution of the usSEABED database used to specify the initial sediment distribution in the model? Why use mud and sand (2 classes) in the large-scale Terrebonne model and mud, silt and sand (3 classes) in the large-scale Atchafalaya model?

The usSEABED provides sampling points data where the percentage of the different fractions are indicted. We took these points and interpolated on the 90 m grid. Most of the sampling points simply separate mud and sand, which is the separation we adopted in the large-scale Terrebonne model. Note that the interpolation approach was already taken by Liu et al. (2018), which modelled the same area. The Atchafalaya model was subsequently and separately developed from the Terrebonne one, and we simply took a different approach. In this area, there is a much smaller number of samples, thus we retrieved an average fraction weight (in percentage) and set a uniform initial concentration. From the measurements we got that the 22% is sand and 78% is mud. The 78% was then divided in half between clay and silt. In this case, we performed a spin-up of the bed by running a one-year simulation with a morphological factor of 50 to reach equilibrium.

Figure 2: Perhaps the caption should refer to Figure 3 for the spatial relationship among the 3 imaged regions.

Yes, this could help the reader better locating the flight lines. We added the reference in the caption: "*Refer to Figure 3 for the spatial relationship*".

L172-176: What value was used for the erosion parameter? Why wasn't this parameter varied in the calibration runs? Its effect is different from the critical shear stress.

The erosion parameter was fixed at 0.00001 kg m$^{-2}$s$^{-1}$. We agree with the reviewer. The erosion parameter has its own effect in the Partheniades-Krone formulation (Partheniades, 1965). In this case, we did not consider variability due to the fact that, despite having two classes of sediment, most of the bed is composed by mud (sand is located more of shore along the barrier island). The value is within the typical range suggested by Winterwerp et al. (2012). The same authors highlight the fact this parameter is more or less constant in the case of homogeneous beds. We acknowledge that this is a limitation. The bed is unlikely that homogeneous and some grain size variability should be expected, thus the erosion parameter could be different from what we assumed. The one selected is a very common value within the range 1-5·10$^{-5}$ kg m$^{-2}$s$^{-1}$ for modeling studies of coastal bays (e.g.Ganju and Schoellhamer (2010); van der Wegen et al., (2011); Wiberg et al., (2015)). We added this information into the methods section.

L178: CRMS station 421 is referred to in this paragraph, but the CRMS acronym is not introduced until the end of section 2 (L226).

We corrected an introduced the acronym earlier when we mention station 421 in the small-scale model description: "*Delft3D was utilized to simulate the hydrodynamics in one of the Delta-X intensive field study sites near station 421 of the Coastwide Reference Monitoring System (CRMS) network (blue rectangle in Figure 1) from 25 March 2021 to 18 April 2021*".

L182: Here and elsewhere, rather than refer to the bottom and upper boundaries of the model grid it would be better to use terms south and north boundaries.

We corrected the inaccuracy everywhere and used the term north and south as suggested.

L183: Where is Trouble Bayou and how far is that from CRMS station 421?

Trouble Bayou is located about 5 km north of station 421 and about 3 km north of the northern boundary of the small-scale model. We added this information in the text: "*The former were the water levels measured by the CRMS site 421, while the latter were computed using water levels measured by CRMS at site 421 and Trouble Bayou located about 5 km north of station 421*".

L186: Why is the Chezy coefficient for the channels set here (55) while the Chezy coefficient for the channels in the large-scale Terrebonne model setup allowed to vary (between 40-45) and over a smaller range of values?

This was a limitation in the large-scale Terrebonne model in the previous version of the manuscript. We now explore a wider range (the same as the Atchafalaya model) and found better results. Now, like in the Atchafalaya model, the best Chezy coefficient is 65 $m^{1/2}s^{-1}$. This is still different from the 55 $m^{1/2}s^{-1}$ used in the small-scale model, however differences between 55 and 65 $m^{1/2}s^{-1}$ are minimal, thus the slightly different values do not affect the results in the small-scale model.

L189-190: Couldn't a change in vegetation roughness affect the water-level changes in addition to marsh topography?

Yes. This an important point raised by another reviewer. In this case the Chezy coefficient does not account for vegetation but only the bottom of the marsh. The 35 $m^{1/2}s^{-1}$ is a typical value used to represent bottom roughness in modelling studies (e.g. Zhang et al., 2018; Passeri et al., 2018). In order to include vegetation a more correct value would range between 10 and 20 $m^{1/2}s^{-1}$ (e.g. Augustijn et al., 2008; Stark et al., 2015) or implement formulation such as Baptist (2005). In this case, our objective was to calibrate elevation and not friction. Calibrating only friction instead of wetland elevation would have led to unrealistic spatial distribution and values of Chezy (Zhang et al., 2022). Indeed, the authors suggest to first calibrate boundary conditions and elevation. Only after these steps, a calibration of the friction would provide more realistic values of friction. For this reason, we simply set an initial and homogeneous friction coefficient. Note that, the calibration of the elevation inherently contains information of vegetation, however, when Zhang et al. (2022) ran a sensitivity analysis on marsh Chezy coefficient found non-significant variation of model performance for all Chezy values (range 8-40 $m^{1/2}s^{-1}$).

Table 1: It could be helpful to provide values of the calibrated parameters (where appropriate) in this table.

Instead of adding the calibrated values in Table 1, we created a second table (Table 2 in the revised manuscript), where we report the best set of parameters for the two large-scale models.

Figure 3: It would be helpful if boxes were used to show the domains of the 3 models in relation to the remote sensing.

We modified Figure 3 and added the boundaries of the three models.

L214: "and the corresponding remote sensing …"

Corrected.

L216: "allow us to tune model parameters …"

For this paper, we decided to use the impersonal sentence structure. Thus, in this case we prefer to keep 'allowed to'.

L241: Delete "For the" before "Atchafalaya model results …"

Corrected.

L246: Why was a Chezy coefficient of 65 not considered in the Terrebonne model? Are any time series available for comparison with the model?

As mentioned in a previous answer, this was a limitation of the model. Now, we have explored a wider range and found that result improve when 65 $m^{1/2}s^{-1}$ is used. We have also added a separate calibration with timeseries and validation using a different period.

L252-3: "water level change in the southern area"?

Yes. Using 'southern' is more appropriate. We corrected.

Fig. 7: It would be helpful to add the date of the image in the figure (as in, e.g., Fig, 8) or in the caption. For the model calibration runs shown in this figure, what settling velocity was used for the results shown in Fig. 7A and what critical shear stress was used for the results shown in Fig. 7B?

We modified Figure 7 accordingly. We added the date and time of the AVIRIS-NG flight. The settling velocity used in Figure 7A, and critical shear stress used in Figure 7B were noted in the caption. However, we added the indication in the figure, so that it is easier for the reader to get the information.

L271: Add a reference to Fig. 8 after noting the date/time of the imagery.

Reference added.

L274: Add reference to Fig. 8B after providing the RMSE of 35.88. Are there any in situ measurements of SSC available for checking model values?

Reference added. Yes, we have available in-situ TSS measurements. We will use them to perform a second calibration that we will compare to the AVIRIS-NG one using a validation period (Delta-X Fall campaign).

Section 4.1-4.3: Are there any reasons to think that the Chezy coefficient might not be uniform?

It is very likely that the Chezy coefficient is not uniform. The area is characterized by patched of *Spartina alterniflora,* thus is it likely to have variable friction. In our case, we are more interested in calibrating the friction. Zhang et al. (2022) suggest to first calibrating the topography using an initial uniform friction. Then calibrating the friction to account for vegetation. They suggest that because a direct calibration of the friction would lead to unrealistic distribution and values. Thus, for the purposes of our analysis, using a uniform friction is admissible. We specified this in the discussion of the UAVSAR coupling results.

L285: Does "lower half of the flight line" = "southern half of the flight line"? If so, it would be clearer to refer to southern half.

Yes. We clarified the expression using southern half.

L285-290: Could the comparison with Manning values be included in results instead of discussion, or is the Manning value so well known that the agreement is a useful point of discussion?

We moved the comparison with Manning in the results.

L306: "The validation of the water levels across the domain …further confirms the goodness of the calibrated friction coefficient" seems like an overstatement given that the previous paragraphs described portion of the model domain where the modeled and remotely sensed water levels do not agree – for reasons that may be unrelated to friction coefficient, but the disagreement makes it impossible to evaluate how well the friction coefficient works in those regions.

In the previous version of the manuscript we were not precise enough. In the new version we have added a new calibration using both AVIRIS-NG and timeseries from gauges, from both calibration and a validation in a different period of the year we show that AirSWOT lead to a better result. The tidal gauges are spread over the domain. We will modify the text and be more precise. We will specify that the validation showed that AirSWOT lead to better results, but there are areas where it is not possible to make this evaluation due to the limitations imposed by the model resolution

L316-327: The argument that friction plays w marginal role in affecting water levels on marsh platforms merits more discussion, since otherwise that seems like a reasonable alternative to arguing that marsh platform topography is poorly quantified. Is there a correlation between where the topography must be adjusted and vegetation characteristics on the marsh? Is it also possible that the model isn't resolving microtopography that affects water fluxes? How much was the topography altered to improve fluxes. Were the values realistic?

This is an excellent point that was not discussed in the previous version of the manuscript. The statement about the marginal role of the friction was not justified. This is a point that was tackled by Zhang et al. (2022) when the method was developed. The calibration of the topography inherently carries information on the friction. The authors ran a sensitivity analysis on the friction coefficient by testing different values (range 8-40 $m^{1/2}s^{-1}$), showing that a variation in friction has no effect on model performance. They point out that it is necessary to first adjust the topography. In the hypothesis of applying the calibration with UAVSAR only on friction, the final friction map would present unrealistic distribution and values because. Thus, they recommend to first calibrate the elevation and then the friction. We added this information in the discussion: "*As suggested by Zhang et al., 2022, friction plays a marginal role in affecting water levels on marsh platforms. They run a sensitivity analysis on the effect of variable friction by exploring a wide range of values*

*finding little effect on model performance. The calibration of the topography inherently contains information of the friction, which can lead to its smaller effect on the computed flow. At the same time, applying the same procedure to the friction (without modifying the marsh elevation) would lead to unrealistic large spatial variation of the friction coefficient, with some unrealistic values*". The modification of the topography produced realistic values. On average it was necessary to decrease the elevation by 1.3 cm with a standard deviation of 16.3 cm. Regarding the point on the microtopography effect on water fluxes, we added additional information, as it was worth of discussion. The method depends on the ability of the model to solve the water fluxes. Therefore, if some areas are not well solved, it could be that the topography is changed even if it is not necessary. For instance, let's suppose there is a small channel that the 10 m mesh cannot represent. During falling tides, the areas near the channel drain faster, hence, they generate a larger water level change compared to internal areas. Since the model cannot solve the channel, the procedure will deepen the topography to allow water to flow and match the water level change, even if the topography is correct. We added this discussion by leveraging on some RTK measurements that we used to validate the calibrated topography: "*this iterative procedure might introduce errors in areas where the topography is correct. This effect can be noted for the three points before mentioned where the marsh is deepened. The methodology depends on how well the model is able to solve all channels. In this particular case, the points are located in proximity to a narrow channel with a 1.5-2 m cross section. The 10 m resolution represents a limitation because features such as channels and levees that are smaller than 10 m cannot be captured by the mesh. Yet, they affect water flow. In this example, we used a UAVSAR captured during falling tide, and in this phase, areas of the marsh close to these channels drains water faster than areas located internally. Since the model does not capture these channels, the method tries to compensate by lowering the marsh to achieve the measured water level change even if the elevation is correct*".

L333: change "worst" to "worse"

Corrected.

L337: Are any data available to evaluate this possibility?

Yes, there are other data available since there were repeat passes of the same flight line. Thus, with an extended analysis this aspect could be evaluated. Certainly, this is an interesting analysis that could be done in the future. For instance, it would be interesting to compare if the bathymetric correction changes (and how much) depending on the use of water level change during rising and falling tides. However, this goes beyond our scope, thus it is not included in the present analysis.

L349: Was TSS sampled just one time and at one location? If so, this doesn't seem like a robust enough test to declare the in situ sample to be more in error than the modeled value. What is the TSS value at that time and place in the AVIRIS-NG data? How much spatial and temporal variability do the model and remote sensing suggest?

There are many samples of TSS done at different times and well distributed in the domain. In order to overcome this limitation, we will perform a second calibration using only in-situ point measurements. Then, we will compare the two calibrations in a validation period (Delta-X Fall campaign). In this way, we will be able to answer the questions and provide a more robust evaluation of the calibration performed with AVIRIS-NG.

L356: Consider revising to "due to flocculation which increases settling velocities compare to …" In any case, the settling velocities used in the model for mud are effectively floc settling velocities.

We revised the sentence. It is correct, in the model these are flocs parameters. Indeed, our goal with the sentence is to remark that the process of flocculation introduces high uncertainty in the settling velocity and critical shear stress because we do not deal with single particles. The emphasis is on the flocs values. We recognized that in the previous version of the manuscript this was not clear. The sentence now reads: "*Especially in coastal areas, cohesive sediment properties are highly affected by flocculation. In this process, fine sediment aggregate to form flocs, for which both settling velocity and bed shear strength of cohesive particles are highly uncertain and difficult to predict*".

L360: Not clear what is meant by: "errors might be related to some the bathymetric modification"

This error is likely related to previous modifications of the text that were not corrected. We eliminated the 'the'. The sentence now reads: "*errors might be related to some bathymetric modification*".

L361: The channels were enlarged in the model? "might have generated"

Corrected.

L363: "inherent to the model. Although Delft-3D has 3-dimensional …"

Corrected.

L365-366: replace "tri-dimensional" with 3D and "bi-dimensional" with 2D.

We corrected all instances of 'tri-'and 'bi-dimensional' with 3D and 2D respectively.

L371: This paragraph might be better combined with the one before.

We agree with the suggestion as the two paragraphs are related. We merged them.

Fig. 9: It would be much better to use the same vertical scale for both profiles. Were the wave conditions much higher on Aug 19 (red curve) than on Aug 17? It seems important to recognize here that optical properties of sediment in suspension is strongly controlled by sediment size, and that the vertical profile of the finest, most optically active fractions might be more uniform that that of coarser bed fractions.

In the first version of Figure 9, we tried to use the same scale. However, the values of water depth (y-axis) have very different ranges. For instance, if we use the red curve range (0-3 m), the blue curve would be squeezed at the bottom of the figure due to its 0-0.5 m range. For this reason, we opted for a double y-axis.

Regarding wave conditions, these were wind speed and direction in the two sampling days:

- 17 August 2021 18:00 UTC: 1.25 m/s 146 degrees north
- 19 August 2021 18:00 UTC: 2.03 m/s 108 degrees north

The sampling point is located within the small-scale model. We do not have measurements of wave conditions in the sampling point (see red dot in Figure 3). Thus, we looked at model results from the large-scale Terrebonne model. Wave conditions were higher on 19 August, but with very small

wave height (Figure 1) and orbital velocity near bottom. This seems to indicate that waves do not have an effect on the differences in sediment profiles.

[Figure]

**Figure 1.** Modelled wave significant height on 17 and 19 August 18:00 UTC. The red rectangle identifies the small-scale domain.

However, there are differences in bottom shear stress, which is likely connected to different water velocity (see Figures 2 and 3). On 19 August both depth-averaged water velocity and bed shear stress are much higher. Interestingly, if we observe the sediment concentration profiles (Figure 4), we can see that only on 19 August the coarser fraction on bed sediment is entrained in the water column, while on 17 August only the finer fraction is measured.

[Figure]

**Figure 2**. Modelled bed shear stress on 17 and 19 August 18:00 UTC. The red rectangle identifies the small-scale domain.

[Figure]

**Figure 3**. Modelled bed shear stress on 17 and 19 August 18:00 UTC. The red rectangle identifies the small-scale domain. The red dot the sampling point.

The reviewer raises an excellent point. We will add the importance of sediment size in the discussion of the AVIRIS-NG results. We will also modify Figure 9 using the figure below (Figure 4 in this document), to make a stronger case.

[Figure]

**Figure 4.** Total sediment concentration profile on the left on 17 and 19 August 2021. On the right the distribution of grain sizes at each depth for both days.

**References**

Feng, Z., Tan, G., Xia, J., Shu, C., Chen, P., & Yi, R. (2020). Two-dimensional numerical simulation of sediment transport using improved critical shear stress methods. *International Journal of Sediment Research*, *35*(1), 15-26.

Ganju, N. K. and Schoellhamer, D. H.: Decadal-timescale estuarine geomorphic change under future scenarios of climate and sediment supply, Estuaries and Coasts, 33, 15–29, https://doi.org/https://doi.org/10.1007/s12237-009-9244-y, 2010.

Liu, K., Chen, Q., Hu, K., Xu, K., and Twilley, R. R.: Modeling hurricane-induced wetland-bay and bay-shelf sediment fluxes, Coastal Engineering, 135, 77–90, https://doi.org/10.1016/j.coastaleng.2017.12.014, 2018.

Partheniades, E.: Erosion and deposition of cohesive soils, Journal of the Hydraulics Division, 91, 105–139, https://doi.org/10.1061/JYCEAJ.0001165, 1965.

Wiberg, P. L., Carr, J. A., Safak, I., and Anutaliya, A.: Quantifying the distribution and influence of non-uniform bed properties in shallow coastal bays, Limnology and Oceanography: Methods, 13, 746–762, https://doi.org/https://doi.org/10.1002/lom3.10063, 2015.

van der Wegen, M., A. Dastgheib, B. E. Jaffe, and J. A. Roelvink. 2011. Bed composition generation for mor-phodynamic modeling: Case study of San Pablo Bay in California, U.S.A. Ocean Dyn.61: 173–186.

Winterwerp, J. C., Van Kesteren, W. G. M., Van Prooijen, B., & Jacobs, W. (2012). A conceptual framework for shear flow–induced erosion of soft cohesive sediment beds. *Journal of Geophysical Research: Oceans*, *117*(C10).

Zhang, X., Jones, C. E., Oliver-Cabrera, T., Simard, M., and Fagherazzi, S.: Using rapid repeat SAR interferometry to improve hydrodynamic models of flood propagation in coastal wetlands, Advances in Water Resources, 159, 104088, https://doi.org/10.1016/j.advwatres.2021.104088, 2022a.

---

## Author Response (AR1)

**Comment on bg-2023-108**

**Co-editor-in-chief decision: Reconsider after major revisions**

**Comments to the author**:
Dear Dr. Cortese and others:

I agree with the guest editors' assessment and invite you to submit a revised version. Thanks for considering Biogeosciences as outlet.

With best regards,

Jack Middelburg, co-ordinating Editor

**Associate editor decision: Reconsider after major revisions**

**Public justification (visible to the public if the article is accepted and published)**:
Dear authors,
Thank you for the detailed reply to the reviewers' comments. I kindly invite you to submit the revised version of the manuscript reflecting the aforementioned changes.
Best regards,
Beatrice

We would like to thank the Co-editor-in-chief and the Associate editor for handling our manuscript. All comments from the reviewers were valuable to improve and ensure quality of our study. In the next lines we address all comments raised by the reviewers. Since many comments were already fully addressed and modifications were already made during the replying process, many answers will not change. In the marked-up version of the reviewed manuscript and supplementary information the new text is written in blue, while the deleted text is crossed and red.

Here we briefly summarize the **major changes** to the manuscript:

- Following Reviewers feedbacks, we added a separate calibration of water levels and sediment properties in the large-scale Terrebonne model using in-situ measurements to compare with the remote-sensing based calibration.
- The calibration with AirSWOT is now more consistent between models, as we explore the same friction values.
- Most of the figures were edited and improved to ensure better readability. In particular, figures for the large-scale Terrebonne model are now focused on the most interesting area and colorbars range narrowed.
- The AVIRIS-NG Figures 7 and 8 were both divided into two figures. In the revised version of the manuscript Figures 7 and 8 are the comparison with AVIRIS-NG in the Terrebonne large-scale model, while Figures 9 and 10 are the comparison with AVIRIS-NG in the Atchafalaya model. This was done to improve clarity of the maps. Note also that Figures 7 and 8 show less cases so that maps can be larger.
- Figure 9 was removed from the manuscript. This figure was used simply to illustrate that sediment concentration can be variable in the water column. This is something that has been measured many times before, thus we decided the simply discuss the effect of sediment heterogeneity. All the conclusions and discussion points raised hold without the need of the figure.

- We improved the description of the UAVSAR-based calibration and added all models' parameters and details that we did not previously include.

**Reply to RC1**

The paper discusses the use of 3 different remote sensing products for the calibration and validation of 3 hydrodynamic models of different scales: AirSWOT to calibrate friction coefficient, UAVSAR to calibrate errors in the intertidal marsh topography and AVIRIS-NG to calibrate sediment properties.

The authors introduce the traditional calibration techniques for such a model, being the use of time series of observations (e.g. water levels, sediment concentration, etc.) and emphasise how remote sensing observations can complement such technique. I believe they innovative character of the research is clearly presented.

We want to thank the Reviewer for providing feedbacks and comments, which greatly helped better presenting our findings. In the following lines, we reply (in **black**) to each comment (in **blue**) and refer to the changes in the manuscript. Modifications are reported between "" and with *italicized* text. At the end, we also report all references to papers cited in the answers.

**Major comments:**

Regarding the use of the AirSWOT data: both the calibration and validation are clearly presented. However, whether the use of AirSWOT actually leads to more accurate model results than calibration using time series is less obvious. Can you calibrate the model using the airswot and using time-series to show in a validation period that the airswot actually appears to be better? In the discussion, could you compare your evaluation statistics with other similar hydrodynamic models of intertidal areas? What are typical RMSE values? If needed, the authors could also calculate other evaluation statistics such as the Nash & Sutcliffe model efficiency to allow comparison with more existing studies.

We extensively updated this part of the paper. We followed the suggestion and performed two separated calibrations using AirSWOT and timeseries. First, we modified subsection 2.7 to include the new calibration and validation (for both water levels and sediment): "*In the large-scale Terrebonne model, a second calibration of the Chezy coefficient and sediment properties was performed using only in-situ observations. Simulated water levels during the Spring campaign were compared with timeseries of 13 tidal gauges within the CRMS network (Figure S3). The RMSE and Nash-Sutclife Model Efficiency (ME) (Allen et al., 2007) were used to evaluate model performance for the different friction coefficients. The validation with timeseries allowed to evaluate the temporal coherence of the results that cannot be captured by remote sensing. The two calibrations were validated in the Fall campaign by comparing water levels. In the same model, a calibration of sediment parameters using in-situ measurements was carried out. Measured TSS concentrations from Fichot et al. (2022) were compared to simulation results (see Figure S5). Finally, the two calibrations were validated with in-situ TSS data collected during the Fall campaign Fichot et al., 2022). RMSE was used to compare model results*".

By doing these modifications we also realized that we could achieve better performance using a friction coefficient of 65 $m^{1/2}s^{-1}$. In the new version of the manuscript, we use the same ranges evaluated in the Atchafalaya model and compare results between 45 and 65 $m^{1/2}s^{-1}$. By doing so, we were also able to detect a few small calculation errors we previously made. Interestingly, we see that in the calibration with AirSWOT, 65 $m^{1/2}s^{-1}$ is the best value, while in the timeseries calibration

we have similar results but slightly better in the 45 $m^{1/2}s^{-1}$ case. We then use the two coefficients in the Delta-X Fall campaign to validate the water levels and find that 65 $m^{1/2}s^{-1}$ works better by comparing water levels gauges. We used the RMSE and Model Efficiency index as suggested to compare performance. We added a new figure in the supplementary material showing the results of the second calibration and the validation. Note also that Figure 4 now shows only the comparison for one moment. In the previous version there were included four different moments. We decided to do this in order to make the image clearer. (in the supplementary there is anyway a figure with all four flight lines). We added a paragraph in the results section: "*The calibrated value of 65 $m^{1/2}$ $s^{-1}$ for the Chezy coefficient is comparable to Manning values for the area. If we consider an average modelled water depth of 1.34 m during the AirSWOT acquisitions and use the water depth as an approximation for hydraulic radius, it can be estimated a correspondent Manning coefficient n = $H^{1/6}$ $C_h^{-1}$ (Limerinos, 1970). The equation yields n = 0.02 $m^{1/3}$ $s^{-1}$ which is within the range of values suggested by the 1992 NLCD and LA-GAP classification (Bunya et al., 2010) for the open water class*".

In the discussion, we added a few comparisons with existing studies to confirm the goodness of our results: "*The results are comparable to other modeling studies at the same location and in similar intertidal areas along the US Atlantic coast. For instance, Mariotti et al. (2010) and Palazzoli et al. (2020) developed models for the shallow coastal bays of the Virginia Coastal Reserve and obtained RMSE of 0.07-0.11 m and 0.26 m respectively. In Louisiana, Freeman et al. (2015) modelled the impact of Hurricane Rita in one of the major coastal lakes in Terrebonne Bay obtaining a RMSE of 0.1 m, whereas Ou et al. (2020) developed a salinity model in the near Barataria Bay and found a RMSE between 0.05 and 0.14 m for water levels*".

UAVSAR: The method is only very briefly explained how the use of water surface elevation changes can be used to calibrate errors in the marsh topography and the authors refer to another paper where more details can be found. As this is an essential part of the paper, I would like to suggest a slightly more extensive explanation of this method. Furthermore, could you address how the UAVSAR - topography calibration can be validated and why no direct validation of the calibrated parameter (being the marsh topography) is included in the paper? Finally, assuming a uniform friction coefficient is very likely not to be the case in reality and how do you ensure that the calibration of the marsh topography does not try to resolve these types of errors instead of errors in the marsh topography.

We agree on the fact that the procedure is important and with the suggestion of providing more details instead of referring to the paper that developed the method. To address this, instead of presenting the principle behind the method, we now explicitly explain the iterative procedure, so that it is more understandable to the reader. Furthermore, we added a new figure in the Supplementary Material, in which we provide a simple flow diagram showing the method (the figure is adapted from Zhang et al., (2022a)). The new method description reads: "*To correct marsh topography, the methodology proposed by Zhang et al. (2022) was followed, in which errors in elevation were corrected by comparing modelled water-level changes with those observed via UAVSAR (see Figure S1). In this procedure, the first simulation is run using the original topography and the difference between modelled and observed water level change is computed in each cell. If the modelled water level change is larger than UAVSAR, the elevation of the cell is increased. In the opposite case, the elevation is decreased. The updated topography is used in the subsequent simulation. The procedure is run with the updated topography iteratively until the minimum RMSE is reached*".

One way to validate the modified bathymetry could be comparing the water level change in a different time using a different UAVSAR flight line. More specifically, one could consider a water

level change measured between two different times. In our case, we chose a change in 3.5 hours. In the description of the procedure, Zhang et al. (2022a) point out that as long as the method converges to a unique solution with realistic values, using a different time would lead to similar results. Another way to validate is considering ground-truth values of elevation measured in-situ. In previous version of the manuscript we did not include any validation, however, in the new version we provide a validation of the results using RTK measurements collected during the Delta-X Spring 2021 campaign and RTK measurements collected by the CRMS. Despite some points show that the original topography was closer to the real value, the coupling with UAVSAR allowed us to reduce the error. It must be noted that the measurements cover a very limited area of the domain. A better validation would consider more spatially distributed point; however, these are not available. We added a sentence in the subsection 2.5: "*To validate the topographic correction in the small-scale model, the original and calibrated topography were compared with Real Time Kinematics elevation measurements collected during the Delta-X Spring 2021 campaign (Twilley and Rovai, 2022) and site 421 elevation provided by the CRMS*". We decided to show the validation figure in the Supplementary Material (Figure S5).

Regarding the friction we added a more detailed point in the discussion in the manuscript. Drag exerted by bottom and vegetation effects the flow. Zhang et al. (2022) showed that model performance did not significantly change for different Chezy coefficients, showing that elevation in these areas exerts a strong influence. They point out, as the Reviewer suggests, that the correction of the topography contains information of the friction. Moreover, they observed that if one would run the calibration process only on friction, this would lead to unrealistic values of frictions and unrealistic spatial distribution. Thus, a good practise would be to set an initial friction, calibrate the topography, and then adjust for friction by paying attention to stay reasonable ranges. In this case, we did not calibrated friction. We added a few sentences in the discussion of UAVSAR: "*As suggested by Zhang et al. (2022), friction plays a marginal role in affecting water levels on marsh platforms. They run a sensitivity analysis with a wide range of friction values and found little effect on model performance. The calibration of topography inherently contains information on friction, which can lead to its small effect on the computed flow field. Applying the same iterative method to friction only, without modifying marsh elevation, would lead to unrealistically large spatial variations of the friction coefficient. Therefore, it was decided to only change marsh elevation to match modelled water-level variations with those derived via UAVSAR*".

AVIRIS-NG: While the calibration process is clearly explained, I believe more emphasis should be put on validating whether the use of remote sensing indeed improves the model over the use of single point data. The author mentions in-situ observations. I would suggest calibrating the model using the in-situ measurements and independently calibrate the model with a AVIRIS-NG image. Then, independent validation (based on either in-situ measurements and/or AVIRIS-NG imagery of a different time period) could indicate whether the model calibrated with the AVIRIS-NG image indeed performs better.

We agree with the evaluation of the reviewer. Now the evaluation of AVIRIS-NG compared to in-situ measurements is more complete. As reported before, we modified subsection 2.7 to include the additional calibrations (sediment included). In the supplementary material we added three new figures. Figure S6 shows the location of the samples for both calibration and validation (locations are different). Figure S7 shows the calibration with only in-situ measurements. Figure S8 shows the comparison of the two separate calibrations in the validation period (Fall 2021 campaign). We found that the calibration provided two different results, while the validation confirmed that the best set is the one found with AVIRIS-NG. In the results section we added a paragraph that reads: "*Settling velocity calibrated using only in-situ measurements was found to be 0.325 mm/s, different from the value obtained from the remote sensing images (Figure S7). The optimal critical shear*

*stress was identical (0.1 Pa). This combination provided the lowest RMSE of 25.26 mg/L. During the Fall campaign, when the two calibrations were compared, the calibration performed using AVIRIS-NG provided better agreement with in-situ measurements than the calibration with in-situ measurements only (Figure S8)*".

In the discussion we eliminated the paragraph where we discussed the two single points and added a new one based on these results: "*A calibration using only measurements in the field presents limitations. The optimal set of parameters is $w_s$ = 0.325 mm/s, $\tau_{cr,e}$ = 0.1 Pa (Figure S6h), which differs from the best set found using AVIRIS-NG ($w_s$ = 0.25 mm/s, $\tau_{cr,e}$ = 0.1 Pa). Interestingly, the validation shows that, despite the model tends to underestimate concentrations in both cases, the calibration with AVIRIS-NG provides the lowest RMSE (Figure S8). AVIRIS-NG provides the possibility to compare the results with spatially distributed data, allowing a more complete evaluation of model performance. Point measurements could also not cover the full range of concentration and lead to calibrated parameters not representative for the entire area. Especially in coastal areas, cohesive sediment properties are highly affected by flocculation. Fine sediment aggregate to form flocs, for which both settling velocity and bed shear strength are highly uncertain and difficult to predict*".

**Minor comments:**

1. Line 134: Could you add the uncertainty on water level measurements.

   For each flight line there is an error file (named *err* in the Denbina et al. (2022) dataset) which contains an estimate of the vertical error for each pixel and provides a spatially-varying estimates based on the interferometric correlation. For both Delta-X campaigns, AirSWOT was validated using in-situ gauges and it was found a Root Mean Squared Error of 9 cm across the entire campaigns when AirSWOT was averaged over a 1 km$^2$ area. We added the last information in the manuscript at the end of the AirSWOT data presentation: "*Water surface elevation data were validated in both Delta-X campaigns using in-situ gauges and a root mean squared error of 9 cm was found when data were averaged on a 1 km$^2$ area*".

2. Line 108: Could you explain why small-scale Terrebonne model domain was chosen as a small-scale region of interest? Could you support that this is a representative area?

   Since the large-scale Terrebonne model covers a very large area (about 90 km east to west), it has a coarse resolution in order to be more computationally efficient. The downside is the inability to reproduce small features. Given the microtidal range, differences in water level change on the marsh depends on small topographic differences, that such a coarse grid cannot capture. Hence, the small-scale model allowed us to incorporate UAVSAR data in our numerical modelling framework. As we mentioned at the beginning of paragraph 2.1, the Terrebonne Basin has been constantly losing marshland. The small-scale domain is located in these degrading salt marshes. Thus, the processes in these areas can be considered representative for most of the Terrebonne Basin.

3. Line 164: could you support the decision for the chosen Chezy coefficients for ocean and marsh platform and on line 186 could you explain why these values differ from the large-scale Terrebone model? Same comment for the values mentioned in line 206.

   The Chezy coefficient range is chosen based on different modelling studies of coastal marshes and deltas. We forgot to add those references. We added a new sentence indicating this: "*The selected values fall within a range considered by several modelling studies of coastal marshes*

*and deltas (e.g. Edmonds and Slingerland (2010); Nardin et al. (2013); Stark et al. (2015); Zhang et al. (2019))".* This applies for both Terrebonne and Atchafalya.
In the new version of the manuscript we improved the AirSWOT analysis and now we explore the same friction range in the two large-scale models. A different Chezy coefficient in the channels in the small-scale model in Terrebonne was used simply because the two models were developed at different time. However, the differences between using 55 or 65 $m^{1/2}s^{-1}$ are minimal and do not significantly affect model results.

1. Figure 4 & Figure 7: Spatial patterns are difficult to observe, could you make the map larger (for instance, by rotating the map so that the flight line is either horizontal or vertical)? Could you change the colour scale to stretch the values in the raster map? It seems both negative and positive values on the difference maps never - and + 0.75 m.

   We have modified both figures to better observe the patterns. We have enlarged the flight line in Figure 4 to focus on the south area of the flight line (similarly to Figure 5). Now the figure is clearer. We have also decided to show only one flight line instead of four, in order to be consistent with Figure 5. We included a figure with the four flight lines in the supplementary material (Figure S2) so the reader can see a comparison of the same flight line in different times of the tidal cycle. We have also reduced the ranges of the error colorbars.

2. Figure 5 and 6: Could you make these maps also bigger? I would propose to only show the colorbar for elevation once and the error colorbar once and if needed, drop the grid labels and add a scale (like figure 4). The scale and zoom of figure 2 is a good example. The use of a single colorbar is shown very well in figure 7. In figure 5, subfigures are not labelled.

   We have modified Figure 5 and Figure 6 and followed the suggestions. Both figures are now larger and more readable. There is only one colorbar for the evelation and error in Figure 5 and only one colorbar for the water level change in Figure 6. In the latter, we also added a land/water mask to better identify channels and marsh and change the order of the subfigures with a better one. We added labels where missing.

3. Line 257: unclear which areas are considered critical and why.

   The critical areas are mentioned in the previous lines: the bottom an upper area. However, it was difficult to connect them to the 'critical areas' later mentioned. Those are considered as such because they show the higher disagreement between simulation and UAVSAR after the first run. We modified the text to correct the unclear sentences. Now the paragraph reads: "*Results from the first run (Figures 6b and 6d) highlight two critical areas. First, the model overestimates the water level change in the southern area, indicating an error in the marsh elevation derived from LiDAR data. The opposite is occurring in the northern area, where waters are found to recede too slowly and consequently the water level change is underestimated*".

4. Line 309: could you refer to the specific figure which supports this statement.

   We forgot to refer to the Figures in the Supplementary Information. In this case the correct reference would have been Figures S1 and S2. However, in the new version of the manuscript we removed these figures to show the results of the calibration with timeseries and validation of the calibrated Chezy in a different period. Now in subsection 3.1, we refer to Figure S5: "*Figures S5a and S5b show the results of the calibration using timeseries of stationary gauges. The calibration shows similar results, with $C_h = 45$ $m^{1/2}s^{-1}$ providing the better RMSE. The validation of the friction coefficient during the Fall campaign (Figures S5c and S5d) shows*

*better performance for $C_h$ = 65 $m^{1/2}s^{-1}$, with a RMSE of 0.047 m compared to the 0.066 m for the 45 $m^{1/2}s^{-1}$ case*".

Corrected. The sentence now reads: "*Here, the model tends to always overestimate sediment concentration and the error decreases as critical shear stress and settling velocity increases*".

5. Line 268: switch order of 'better' and 'performs'.

   Corrected. The sentence now reads: "*Overall, the model performs better in the open sections of the flight line, while at the extremities, performance declines*".

6. Line 269: drop the first 'best' and the sentence misses an active verb (online 2017 change 'as' to 'is').

   Corrected. The sentence now reads: "*The combination that provides the best comparison with measurements is a critical shear stress of 0.1 Pa and a sediment settling velocity of 0.25 mm/s*".

**References**

Denbina, M., Simard, M., and Rodriguez, E.: Delta-X: AirSWOT L2 Geocoded Water Surface Elevation, MRD, Louisiana, 2021, Version 2. ORNL DAAC, Oak Ridge, Tennessee, USA, https://doi.org/10.3334/ORNLDAAC/2128, 2022.

Zhang, X., Jones, C. E., Oliver-Cabrera, T., Simard, M., and Fagherazzi, S.: Using rapid repeat SAR interferometry to improve hydrodynamic models of flood propagation in coastal wetlands, Advances in Water Resources, 159, 104088, https://doi.org/10.1016/j.advwatres.2021.104088, 2022a.

**Reply RC2**

This study tries to improve numerical model predictions and validation of sediment fluxes between tidal marshes, channels and bays, by data assimilation of high resolution remote sensing imagery.

This topic is worthwhile and important to explore, timely and well suited for the audience of biogeosciences.

The study is of high quality and well written.

We want to thank the Reviewer for taking the time to carefully read our manuscript and providing comments to better present our work. In the following lines, we reply (in **black**) to each comment (in **blue**) and refer to the changes in the manuscript. Modifications are reported between "" and with *italicized* text. At the end, we also report all references to papers cited in the answers.

Line 65: could more information been added here. I assume the authors are referring to surface concentrations of suspended solids derived from AVIRIS-NG

Yes, we refer to the **surface** sediment concentration. We modified the sentence to include this information. In addition, we added an information on the method used to derive those data: "*Jensen et al. (2019) used high resolution remote-sensing reflectance data from NASA's Airborne Visible/Infrared Imaging Spectrometer-Next Generation (AVIRIS-NG) to derive maps of total surface suspended solids in the waters of the Atchafalaya basin along the Louisiana coast (USA). They used an algorithm centered on a derivative-based partial least squares regression between measured total surface suspended solids and in-situ spectra*".

Line 95: this might be just a detail but I suggest to put species names in italic

Using italicized text is more precise, so we applied the suggested modification.

Line 125ff: how can the UAVSAR and AirSWOT distinguish between emerged vegetation or the water surface? Are there vertical errors ranges that could be reported?

In UAVSAR, in order to separate wetlands and water surface, a water mask is generated from the interferogram L1 products. AirSWOT mounts a digital camera that maps the water surface. However, in the case of the Delta-X mission the camera wasn't mounted as it requires cloud-free conditions. Since UAVSAR is better at discriminating water and wetlands, the UAVSAR water mask was used also for AirSWOT. We added two sentences specifying this: "*In flooded wetlands, the water surface is detected through the double-bounce scattering mechanism from water and vegetation (Kim et al., 2009; Wdowinski et al., 2013). To separate the water surface from the emergent wetland, a water mask was generated from the interferogram Level-1 products*". In the AirSWOT data introduction we added: "*AirSWOT uses cross-track interferometry to measure the elevation and combines it with along-track interferometry to correct for the bias due to the water motion (Goldstein and Zebker, 1987). To separate land from water surface, the same UAVSAR water mask was used for AirSWOT*".

Regarding the vertical error, there is an error file (named *err* in the Denbina et al. (2022) dataset) which contains an estimate of the vertical error for each pixel and provides a spatially-varying estimates based on the interferometric correlation. For both Delta-X campaigns, AirSWOT was validated using in-situ gauges and found a Root Mean Squared Error of 9 cm across the entire campaigns when AirSWOT was averaged over a 1 km$^2$ area. We added the last information in the manuscript at the end of the AirSWOT data presentation: "*Water surface elevation data were validated in both Delta-X campaigns using in-situ gauges and a root mean squared error of 9 cm was found when data were averaged on a 1 km$^2$ area*".

Line 144ff: is TSS surface or depth averaged concentration derived? How applicable is the calibration during different types of suspended particles during high and low flow conditions?

The TSS derived from AVIRIS-NG imagery is surface concentration. The in-situ measurements used to develop and validate the algorithms were collected in a wide range of contrasting water types in Terrebonne and Atchafalaya Bay during both Delta-X campaign, which covered high flow (during Spring 2021) and low flow (during Fall 2021) conditions. The algorithm to retrieve TSS from AVIRIS-NG performed well (Median Absolute Percent Difference 13.7% and Median bias 6.71 mg L$^{-1}$) across a wide range of TSS concentrations (0.1–154.5 mg L$^{-1}$) throughout both basins and in contrasting seasons. More detailed information on the spatial and temporal coverage of the sampling and algorithm calibration are in Fichot and Herringmeyer (2022) and Fichot and Herringmeyer (2023). We modified the paragraph related to AVIRIS-NG data and included these informations: "*Local empirical algorithms derived using in-situ measurements were used to derive TSS concentration form the Rrs(λ) in the visible/near-infrared region and generate maps of TSS from the AVIRIS-NG imagery (Gao et al., 1993; Bue et al., 2015; Jensen et al., 2019). In-situ samples were collected in both Terrebonne and Atchafalaya basins during the Delta-X 2021 Spring and Fall campaigns in order to capture high and low flow conditions. The algorithm to retrieve TSS from AVIRIS-NG performed well (Median Absolute Percent Difference 13.7\% and Median bias 6.71 mg/l) across a wide range of TSS concentrations (0.1–154.5 mg/l) (Fichot and Herringmeyer, 2021, 2022)*".

Line 159: what was the grain size, what transport equations were used?

We set the sand median diameter at 0.14 mm. For the cohesive fraction, we defined the settling velocity and not the grain size. The sand transport is modelled using Van Rijn (2007), while the mud transport is modelled with Partheniades-Krone (Partheniades, 1965). A few of this information was already included, however a part was not indicated. Now we explicitly state the equation used. *"Thus, the calibration of the parameters refers to the properties of the mud fraction. The default transport equation of Van Rijn (2007) was used for sand". "The default Delft3D sediment transport formulation of Partheniades-Krone (Partheniades, 1965) was used for the non-cohesive fraction".*

How was the bed initialized, 1- or multiple layers?

The bed is initiated as 1 layer. In each cell we defined a percentage of sand and mud based on Williams et al. (2006).

What was the active layer thickness?

The layer thickness was set at 5 m to ensure we did not run out of sediment to resuspend.

Was the option for mixed sediments used?

We did not use the option for mixed sediments and two fractions were modelled separately.

How was the non-cohesive/cohesive boundary defined?

The separation between sand and mud (non-cohesive and cohesive fraction) was based on a previous study in the area (Liu et al., 2018), in which they model deposition generated by Hurricane Gustav. Median sand diameter size is set to 0.14 mm, which locates the sediment in the fine sand class (using the Wentworth scale). Everything below is considered cohesive.

Line 165: is it correct to assume that for tidal channels/lakes and bays the same chezy coefficient was used? did the author try a spatial varying chezy? which wetting and drying scheme was used?

This is a very interesting point that was discussed during the development of the model. The assumption of a homogeneous Chezy coefficient holds some simplifications and initially we tried to use a spatially varying Chezy coefficient to improve the comparison with AirSWOT, especially for the smaller areas and channels within the marsh. However, we realized that for the large-scale models, results did not improve by varying the value of the friction. The limitations of the coarse grid outcompeted those due to the same friction coefficient. Namely, before considering more tuning of the friction, the model must correctly solve the flow in smaller areas and channels. In a possible future improvement of the models, using an unstructured grid could allow better representing smaller features. For instance, a separation in terms of friction could be done between ocean/main tidal channels and small tidal creeks.

The wetting and dry scheme is based on a threshold depth which was set at 1 cm. The algorithm 'activates' a cell when the water depth is positive and larger than the threshold, while the cell becomes dry wet the water depth falls below the threshold. More precisely, the algorithm uses half of the wetting threshold to avoid a "flip-flop" (a change of state in two consecutive time steps, due to oscillation generated by the algorithm). We added a sentence in subsection 2.5: "*A threshold depth of 1 cm was set for the wetting-drying scheme*". We added the sentence here since we focus on marsh flooding only in the small-scale model.

Line 170: I assume ws is calculated from the median diameter and the sediment density I assume the parteniades krone relation is used for sediment pickup please indicate? See comment above what equation was used for sand.

Yes, settling velocity of sand is computed as function of the median diameter and density according to Van Rijn (1993), where one out of three different equations is used depending on the median diameter. We added the information in the manuscript: "*Note that in the case of non-cohesive particles, Delft3D does not require to specify a value for the settling velocity, since it is directly computed from the median diameter and density using the Van Rijn (1993) approach*".
The Partheniades-Krone formulation was used for the cohesive fraction. Please refer to the previous answers for more details and related modification to the manuscript.

Line 175: similar comment as above why was only one muddy sediment class in the bed considered? For instance consolidated clay lenses can possess high crit. bss.,, .. did something like that occur? how well was the initial stratigraphy incorporated in the model?

We acknowledge that within our mud class multiple subclasses are present, thus our separation represents a simplification of the real conditions. The main reasons we opted for a two-class separation, is that the only robust dataset we have to represent the bottom (Williams et al., 2006) provides sampling points where most of the time this is the only separation. As pointed out in an earlier comment, the initial stratigraphy is based on this dataset.

Line 185: is the marsh platform chezy also representing vegetation?

In this case the Chezy coefficient does not account for vegetation but only the bottom of the marsh. The 35 $m^{1/2}s^{-1}$ is a typical value used to represent bottom roughness in modelling studies (e.g. Zhang et al., 2018; Passeri et al., 2018). In order to include vegetation a more correct value would range between 10 and 20 $m^{1/2}s^{-1}$ (e.g. Augustijn et al., 2008; Stark et al., 2015) or implement formulations such as Baptist (2005). In this case, our objective was to calibrate elevation and not friction. Calibrating only friction instead of wetland elevation would have led to unrealistic spatial distribution and values of Chezy (Zhang et al., 2022). Indeed, the authors suggest to first calibrate boundary conditions and elevation. Only after these steps, a calibration of the friction would provide more realistic values of friction. For this reason, we simply set an initial and homogeneous friction coefficient. Note that, the calibration of the elevation inherently contains information of vegetation, however, when Zhang et al. (2022) ran a sensitivity analysis on marsh Chezy coefficient found non-significant variation of model performance for all Chezy values (range 8-40 $m^{1/2}s^{-1}$).

Line 199: what kind of data was provided.. suspended sediment concentration,..?

Yes, it is suspended sediment concentration data. We were not precise, so we add the 'concentration' term in the sentence.

Line 205: was the initial sediment distribution uniform...? was a spinup for the bed tested?

Yes, the Atchafalaya model initial sediment distribution on the bed was taken uniform and it is based on the usSEABED database. A spin-up of the bed is run for about one year with a morphological speeding factor of 50 to reach the equilibrium state. During this process, the bed level is kept fixed, and the bed fraction changes to adapt to the hydrodynamics. The bed composition after the spin-up process becomes more realistic and coarser fractions appear in areas with stronger flow shear stress. We followed an approach consistent with previous studies such as van der Wegen et al. (2011) and Zhang et al. (2020). We have added this information in the

subsection 2.6: "*For sediment transport, three sediment classes were considered: sand, silt, and mud. The initial sediment distribution was derived from the usSEABED database and set uniform at 22% sand, 39% silt, and 39% clay. Settling velocities for silt and clay were fixed at 1 mm/s and 0.5 mm/s, respectively, while the median diameter of sand was set at 0.1 mm. Due to the limited number of samples in the area, a bed spin-up process of one year was run with a morphological speeding factor of 50 to reach bottom equilibrium. During this process, the bed level was kept fixed, and the bed fractions changed to adapt to the hydrodynamics (see similar approaches in van der Wegen et al. (2011) and Zhang et al. (2020))*".

Line 215: see comment above why was no spatial tunning test? this could also improve fig.5?

Regarding the spatial tuning of the friction for Figure 5, first, it is important to mention that the area in dark blue colour on the left side of the flight line is affected by the edge effect. The primary instrument of AirSWOT is the interferometric synthetic aperture radar called KaSPAR. The KaSPAR outer swath mode has incident angle between 4 and 25°. This means that at the edges of this range we find the largest error (Denbina et al., 2019). At the edges the water surface elevation might not be correct. This is something we did not mention in the discussion. We added a new sentence in subsection 4.1: "*It has to be noted that the AirSWOT flight lines presents a large error at the edges (blue areas in Figure 5a) because the incident angle of the outer swath mode is within 4 and 25 degrees (Denbina et al., (2019)*". Another point regarding the friction, is that the friction is typically defined for broad classes, in particular for areas that are constantly underwater such as channels. A reasonable change in friction coefficient (meaning changing the values to physically sound values) would not particularly change the water levels.

Line 240: is the RMSE calculated over an entire M2 tide or only during the time-slice the picture was taken?

The RMSE associated with the remote sensing (the one showed in Figures 4d and 4e for instance) refers only to that particular snapshot. The RMSE used to evaluate the calibration and validation using timeseries is computed separately by including different CRMS water level gauges (see Figure S4).

Fig.7: although the model result are very impressive, the predicted error in SSC is still between 30% - 50% of the measured range, for what conditions in the tide is this representative, is this the best or worst case?

This flight line was collected on 05 April 2021 at 19:57 UTC. This is approximately one hour after low tide when the tide is rising. The choice of the flight line was mostly dictated by the quality of the retrieved TSS. Cloud conditions were optimal in the area covered by the flight line (high quality flight line). Moreover, there is one of the sampling points used to calibrate the algorithm used to retrieve TSS from AVIRIS-NG. Thus, this is one of the best cases we could show of all the available flight lines.

I think fig.7 could be improved since open water or marsh area are difficult to distinguish? If I interpret the results correctly the biggest error seem to appear in shallow areas?

We have improved Figure 7 by making it larger and more readable. We tested if there is a correlation between the error and depth. We found no relationship. There are areas with very similar depth that have different errors. In text, we offer some reason that could explain why we see that. First, we needed to enlarge and deepen some channels to allow a correct tidal propagation. This might have introduced errors in the flow velocities that have a direct effect on the resuspension of

sediments. Second, there are areas with similar depth where the flow is slower and does not resuspend enough. This last point is also related to the lack of heterogeneity in our sediments. It is possible that we do not account for the finest particle that can be resuspended even in weak currents.

Line 295: was a finer mesh tested,.. was an unstructured grid tested?

In very preliminary test we tested a 50 m grid, which did not provide significant improvement but only enhanced computational costs. We did not implement an unstructured grid. This is a limitation of the models, which can reproduce water levels in the wider open areas but struggles to resolve smaller channels. This limitation can be partly addressed by switching to an unstructured grid. It has also to be noted that the unstructured grid would have limitations. Many small channels and creeks are about 1-2 m wide. Such a small size would be challenging also for an unstructured grid.

Line 305: see comment above,.. very low water levels and gradients,.. will make the mini. water depth and flooding and drying scheme relatively important?

The reviewer is correct. As we mentioned in a previous answer the wetting-drying scheme is based on a threshold depth, thus the minimal water depth is very important. Delft3D-FLOW uses an element removal algorithm. For instance, the FVCOM model, which is employed by the Deb et al. (2023) paper introduced in the next question, uses a thin film algorithm (Medeiros and Hagen, 2013). Using a different wetting and drying method would affect the final result, thus it is important to provide the information in the methods section. We added the sentence: "*A threshold depth of 1 cm was set for the wetting-drying scheme*".
It is worth noting that, this observation is only valid for the marsh area that are undergo flooding and drying, therefore only the application of UAVSAR. The other applications are related to channels and bays which are always underwater (no drying involved).

Line 320: how did the marsh microtopography influence the wetting and especially drying of the interior, i.e. Deb 2023,.. used an additional porosity to limit ponding caused by submesh channels?

As we mention in the manuscript, due to the very small tidal range, the topography of the marsh plays a crucial role in the wetting and drying. The suggested paper introduces to a clever way to deal with submesh channels, which is a problem also in our case. The authors modified FVCOM, which is very different from the structured grid version of Delft3D we used. As we said a few answers earlier, this is a limit of the models we developed. The problem of artificial ponding caused by submesh channels that are not solved is indeed a limitation of the methods used to correct the bathymetry. The method depends on the ability of the model to solve the water fluxes. Therefore, if some areas are not well solved, it could be that the topography is changed even if it is not necessary. For instance, let's suppose there is a small channel that the 10 m mesh cannot represent. During falling tides, the areas near the channel drain faster, hence, they generate a larger water level change compared to internal areas. Since the model cannot solve the channel, the procedure will deepen the topography to allow water to flow and match the water level change, even if the topography is correct. We added this discussion by leveraging on some RTK measurements that we used to validate the calibrated topography: "*this iterative procedure might introduce errors in areas where the topography is correct. The methodology also depends on how well the model can solve the tidal channels. This effect can be noted for the three points before mentioned where the marsh was deepened. In this case, the points are located in proximity to a narrow channel with a 1.5-2 m cross section. The 10 m resolution of the model represents a limitation because features such as channels and levees that are smaller than 10 m cannot be captured by the mesh. In this example, the UAVSAR flight line captured the flow during falling tide, and in this phase, areas of the marsh close*

*to the channels drained faster than internal ones. Since the model does not capture these channels, the method tries to compensate by lowering the marsh to increase water fluxes even if the elevation is correct*".

Line 370: it is unclear how the davg sediment concentration predicted by D3d was correct to compare to the remote sensing product.

This is an important point that was also raised by the other reviewers, and we agree that the previous version of the paper does not clearly show it. In the revised version we have added a separate calibration using only in-situ measurements. We have also added a comparison of the two calibrations in a validation period (Delta-X Fall 2021 Campaign). We have added a paragraph in subsection 2.7 to explain the separate calibrations: "*In the large-scale Terrebonne model, a second calibration of the Chezy coefficient and sediment properties was performed using only in-situ observations. Simulated water levels during the Spring campaign were compared with timeseries of 13 tidal gauges within the CRMS network (Figure S3). The RMSE and Nash-Sutclife Model Efficiency (ME) (Allen et al., 2007) were used to evaluate model performance for the different friction coefficients. The validation with timeseries allowed to evaluate the temporal coherence of the results that cannot be captured by remote sensing. The two calibrations were validated in the Fall campaign by comparing water levels. In the same model, a calibration of sediment parameters using in-situ measurements was carried out. Measured TSS concentrations from Fichot et al. (2022) were compared to simulation results (see Figure S5). Finally, the two calibrations were validated with in-situ TSS data collected during the Fall campaign (Fichot et al., 2022). RMSE was used to compare model results*".

We found that the calibration with in-situ measurements provides a different set of best parameters. The validation showed that the calibration with AVIRIS-NG performs better. We added this information in the results section; "*Settling velocity calibrated using only in-situ measurements was found to be 0.325 mm/s, different from the value obtained from the remote sensing images (Figure S7). The optimal critical shear stress was identical (0.1 Pa). This combination provided the lowest RMSE of 25.26 mg/L. During the Fall campaign, when the two calibrations were compared, the calibration performed using AVIRIS-NG provided better agreement with in-situ measurements than the calibration with in-situ measurements only (Figure S8)*". $w_s = 0.325$ mm/s, $\tau_{cr,e} = 0.1$ Pa

We also added discussion points: "*A calibration using only measurements in the field presents limitations. The optimal set of parameters is $w_s = 0.325$ mm/s, $\tau_{cr,e} = 0.1$ Pa (Figure S6h), which differs from the best set found using AVIRIS-NG ($w_s = 0.25$ mm/s, $\tau_{cr,e} = 0.1$ Pa). Interestingly, the validation shows that, despite the model tends to underestimate concentrations in both cases, the calibration with AVIRIS-NG provides the lowest RMSE (Figure S8). AVIRIS-NG provides the possibility to compare the results with spatially distributed data, allowing a more complete evaluation of model performance. Point measurements could also not cover the full range of concentration and lead to calibrated parameters not representative for the entire area. Especially in coastal areas, cohesive sediment properties are highly affected by flocculation. Fine sediment aggregate to form flocs, for which both settling velocity and bed shear strength are highly uncertain and difficult to predict*".

**References**

Augustijn, D. C., Huthoff, F., & Van Velzen, E. H. (2008). Comparison of vegetation roughness descriptions. *Altinakar, MS, Kokpinar, MA, Aydin, I., Cokgor, S. Kirkgoz, S.(eds.) River Flow*, *2008*, 343-350.

Baptist, M.J., 2005. Modelling Floodplain Biogeomorphology. Delft University of Technology, TU Delft.

Denbina, M., Simard, M., and Rodriguez, E.: Delta-X: AirSWOT L2 Geocoded Water Surface Elevation, MRD, Louisiana, 2021, Version 2. ORNL DAAC, Oak Ridge, Tennessee, USA, https://doi.org/10.3334/ORNLDAAC/2128, 2022.

Denbina, M., Simard, M., Rodriguez, E., Wu, X., Chen, A., and Pavelsky, T.: Mapping water surface elevation and slope in the mississippi river delta using the AirSWOT Ka-Band interferometric synthetic aperture radar, Remote Sensing, 11, 2739, https://doi.org/10.3390/rs11232739, 2019.

Fichot, C.G., and J. Harringmeyer. 2022. Delta-X: AVIRIS-NG L3-derived Water Quality, TSS, and Turbidity, MRD, LA 2021, V2. ORNL DAAC, Oak Ridge, Tennessee, USA. https://doi.org/10.3334/ORNLDAAC/2112

Fichot, C.G., and J. Harringmeyer. 2022. Delta-X: In Situ Water Surface Reflectance across MRD, LA, USA, 2021, Version 2. ORNL DAAC, Oak Ridge, Tennessee, USA. https://doi.org/10.3334/ORNLDAAC/2076

Liu, K., Chen, Q., Hu, K., Xu, K., and Twilley, R. R.: Modeling hurricane-induced wetland-bay and bay-shelf sediment fluxes, Coastal Engineering, 135, 77–90, https://doi.org/10.1016/j.coastaleng.2017.12.014, 2018.

Medeiros, S. C., & Hagen, S. C. (2013). Review of wetting and drying algorithms for numerical tidal flow models. *International journal for numerical methods in fluids*, *71*(4), 473-487. https://doi.org/10.1002/fld.3668

Partheniades, E.: Erosion and deposition of cohesive soils, Journal of the Hydraulics Division, 91, 105–139, https://doi.org/10.1061/JYCEAJ.0001165, 1965.

Passeri, D. L., Long, J. W., Plant, N. G., Bilskie, M. V., & Hagen, S. C. (2018). The influence of bed friction variability due to land cover on storm-driven barrier island morphodynamics. *Coastal Engineering*, *132*, 82-94.

Stark, J., Van Oyen, T., Meire, P., & Temmerman, S. (2015). Observations of tidal and storm surge attenuation in a large tidal marsh. *Limnology and Oceanography*, *60*(4), 1371-1381.

Van der Wegen, M., Dastgheib, A., Jaffe, B. E., & Roelvink, D. (2011). Bed composition generation for morphodynamic modeling: Case study of San Pablo Bay in California, USA. Ocean Dynamics, 61(2–3), 173–186. https://doi.org/10.1007/s10236-010-0314-2

Van Rijn, L. C. (2007). Unified view of sediment transport by currents and waves. I: Initiation of motion, bed roughness, and bed-load transport. *Journal of Hydraulic engineering*, *133*(6), 649-667.

Van Rijn, L. C. (1993). Principles of sediment transport in rivers. *Estuaries and Coastal Seas. Aqua Publications*.

Williams, S. J., Arsenault, M. A., Buczkowski, B. J., Reid, J. A., Flocks, J., Kulp, M. A., Penland, S., and Jenkins, C. J.: Surficial sediment character of the Louisiana offshore Continental Shelf region: a GIS Compilation, Tech. rep., US Geological Survey, https://doi.org/10.3133/ofr20061195, 2006.

Zhang, X., Leonardi, N., Donatelli, C., & Fagherazzi, S. (2019). Fate of cohesive sediments in a marsh-dominated estuary. *Advances in water resources*, *125*, 32-40.

Zhang, X., Leonardi, N., Donatelli, C., & Fagherazzi, S. (2020). Divergence of sediment fluxes triggered by sea-level rise will reshape coastal bays. Geophysical Research Letters, 47(13), e2020GL087862

Zhang, X., Jones, C. E., Oliver-Cabrera, T., Simard, M., and Fagherazzi, S.: Using rapid repeat SAR interferometry to improve hydrodynamic models of flood propagation in coastal wetlands, Advances in Water Resources, 159, 104088, https://doi.org/10.1016/j.advwatres.2021.104088, 2022.

**Reply RC3**

I strongly agree with the central argument of this paper, that spatial information from remote sensing provides critical data for calibrating and testing numerical models of coastal wetlands. The paper reports on results of an experiment in which 3 remote sensing products were used in conjunction with modeling of a region of coastal Louisiana to evaluate the value of combining these approaches. Overall I think the analysis is thorough and informative, but the text would benefit from editing for clarity. Most of my comments are editorial.

We want to thank the Reviewer for the positive support to our study. We are grateful for all comments, and editorial suggestions, which allowed us to better show our analysis. In the following lines, we reply (in **black**) to each comment (in **blue**) and refer to the changes in the manuscript. We report the related modification in the manuscript between "" and with *italicized* text. At the end, we also report all references to papers cited in the answers.

Comments:

L18: "Peter Sheng et al." should just be "Sheng et al."

We doubled checked and the correct citation is Peter Sheng (see https://www.nature.com/articles/s41598-022-06850-z#citeas)

L19: There is a recent paper by Temmerman et al (2023; Reviews of Marine Science) that would be appropriate to cite here.

 The suggested paper is very appropriate here, thus we added the citation.

L48: Does Gross Primary Productivity need to be capitalized?

No, it does not. We corrected the mistake.

L60: change to: ", and coastal wetland above-ground biomass …"

Changed.

L106-112: The statement is made here that there are 3 models at different scales, but it seems that the scale of the large-scale Terrebonne model and the Atchafalaya model are essentially the same.

That is corrected. The large-scale Terrebonne model and the Atchafalaya model are developed at basin scale, and they are comparable (they also have the same spatial resolution). The paragraph was not precise, hence we added an additional sentence to clarify: "*Two models were set-up at the basin scale, while the third one was developed at smaller scale*".

Corrected.

L145: Should the phrase "and vegetation structure are produced" be deleted? It seems the same point is made in the next sentence.

Yes. These lines were badly constructed. We reformulated and added more information regarding AVIRIS-NG and the algorithm to retrieve TSS: "*Local empirical algorithms derived using in-situ measurements were used to derive TSS concentration from the Rrs(λ) in the visible/near-infrared region and generate maps of TSS from the AVIRIS-NG imagery (Gao et al., 1993; Bue et al., 2015; Jensen et al., 2019). In-situ samples were collected in both Terrebonne and Atchafalaya basins during the Delta-X 2021 Spring and Fall campaigns in order to capture high and low flow conditions. The algorithm to retrieve TSS from AVIRIS-NG performed well (Median Absolute Percent Difference 13.7\% and Median bias 6.71 mg/l) across a wide range of TSS concentrations (0.1–154.5 mg/l) (Fichot and Harringmeyer, 2021, 2022). AVIRIS-NG images were also used to produce maps of vegetation structure (Jensen et al., 2021)*".

L162: What is the resolution of the usSEABED database used to specify the initial sediment distribution in the model? Why use mud and sand (2 classes) in the large-scale Terrebonne model and mud, silt and sand (3 classes) in the large-scale Atchafalaya model?

The usSEABED provides sampling points data where the percentage of the different fractions are indicted. We took these points and interpolated on the 90 m grid. Most of the sampling points simply separate mud and sand, which is the separation we adopted in the large-scale Terrebonne model. Note that the interpolation approach was already taken by Liu et al. (2018), which modelled the same area. The Atchafalaya model was subsequently and separately developed from the Terrebonne one, and we simply took a different approach. In this area, there is a much smaller number of samples, thus we retrieved an average fraction weight (in percentage) and set a uniform initial concentration. From the measurements we got that the 22% is sand and 78% is mud. The 78% was then divided in half between clay and silt. In this case, we performed a spin-up of the bed by running a one-year simulation with a morphological factor of 50 to reach equilibrium. We added a sentence in the method section: "*Due to the limited number of samples in the area, a bed spin-up process of one year was run with a morphological speeding factor of 50 to reach bottom equilibrium. During this process, the bed level was kept fixed, and the bed fractions changed to adapt to the hydrodynamics (see similar approaches in van der Wegen et al. (2011) and Zhang et al. (2020b)). The bed composition after this spin-up process became realistic and the coarser fractions appeared in areas with strong bottom shear stress*".

Figure 2: Perhaps the caption should refer to Figure 3 for the spatial relationship among the 3 imaged regions.

Yes, this could help the reader better locating the flight lines. We added the reference in the caption: "*Refer to Figure 3 for the spatial relationship*".

L172-176: What value was used for the erosion parameter? Why wasn't this parameter varied in the calibration runs? Its effect is different from the critical shear stress.

The erosion parameter was fixed at 0.00001 kg m$^{-2}$s$^{-1}$. We agree with the reviewer. The erosion parameter has its own effect in the Partheniades-Krone formulation (Partheniades, 1965). In this case, we did not consider variability due to the fact that, despite having two classes of sediment,

most of the bed is composed by mud (sand is located more of shore along the barrier island). The value is within the typical range suggested by Winterwerp et al. (2012). The same authors highlight the fact this parameter is more or less constant in the case of homogeneous beds. We acknowledge that this is a limitation. The bed is unlikely that homogeneous and some grain size variability should be expected, thus the erosion parameter could be different from what we assumed. The one selected is a very common value within the range $1\text{-}5\cdot10^{-5}$ kg m$^{-2}$s$^{-1}$ for modeling studies of coastal bays (e.g.Ganju and Schoellhamer (2010); van der Wegen et al., (2011); Wiberg et al., (2015)). We added this information into the methods section.

L178: CRMS station 421 is referred to in this paragraph, but the CRMS acronym is not introduced until the end of section 2 (L226).

We corrected an introduced the acronym earlier when we mention station 421 in the small-scale model description: "*Delft3D was utilized to simulate the hydrodynamics in one of the Delta-X intensive field study sites near station 421 of the Coastwide Reference Monitoring System (CRMS) network (blue rectangle in Figure 1) from 25 March 2021 to 18 April 2021*".

L182: Here and elsewhere, rather than refer to the bottom and upper boundaries of the model grid it would be better to use terms south and north boundaries.

We corrected the inaccuracy everywhere and used the term north and south as suggested.

L183: Where is Trouble Bayou and how far is that from CRMS station 421?

Trouble Bayou is located about 5 km north of station 421 and about 3 km north of the northern boundary of the small-scale model. We made a small change and used the water levels from the large-scale model as boundary conditions finding no difference. We updated the sentence: "*Water levels were imposed as boundary conditions at the south and north boundaries. Water levels at both boundaries were extracted from the large-scale model in the respective locations*".

L186: Why is the Chezy coefficient for the channels set here (55) while the Chezy coefficient for the channels in the large-scale Terrebonne model setup allowed to vary (between 40-45) and over a smaller range of values?

This was a limitation in the large-scale Terrebonne model in the previous version of the manuscript. We now explore a wider range (the same as the Atchafalaya model) and found better results. Now, like in the Atchafalaya model, the best Chezy coefficient is 65 m$^{1/2}$s$^{-1}$. This is still different from the 55 m$^{1/2}$s$^{-1}$ used in the small-scale model, however differences between 55 and 65 m$^{1/2}$s$^{-1}$ are minimal, thus the slightly different values do not affect the results in the small-scale model.

L189-190: Couldn't a change in vegetation roughness affect the water-level changes in addition to marsh topography?

Yes. This an important point raised by another reviewer. In this case the Chezy coefficient does not account for vegetation but only the bottom of the marsh. The 35 m$^{1/2}$s$^{-1}$ is a typical value used to represent bottom roughness in modelling studies (e.g. Zhang et al., 2018; Passeri et al., 2018). In order to include vegetation a more correct value would range between 10 and 20 m$^{1/2}$s$^{-1}$ (e.g. Augustijn et al., 2008; Stark et al., 2015) or implement formulation such as Baptist (2005). In this case, our objective was to calibrate elevation and not friction. Calibrating only friction instead of wetland elevation would have led to unrealistic spatial distribution and values of Chezy (Zhang et al., 2022). Indeed, the authors suggest to first calibrate boundary conditions and elevation. Only

after these steps, a calibration of the friction would provide more realistic values of friction. For this reason, we simply set an initial and homogeneous friction coefficient. Note that, the calibration of the elevation inherently contains information of vegetation, however, when Zhang et al. (2022) ran a sensitivity analysis on marsh Chezy coefficient found non-significant variation of model performance for all Chezy values (range 8-40 $m^{1/2}s^{-1}$).

Table 1: It could be helpful to provide values of the calibrated parameters (where appropriate) in this table.

Instead of adding the calibrated values in Table 1, we created a second table (Table 2 in the revised manuscript), where we report the best set of parameters for the two large-scale models.

Figure 3: It would be helpful if boxes were used to show the domains of the 3 models in relation to the remote sensing.

We modified Figure 3 and added the boundaries of the three models.

Corrected.

For this paper, we decided to use the impersonal sentence structure. Thus, in this case we prefer to keep 'allowed to'.

L241: Delete "For the" before "Atchafalaya model results …"

Corrected.

L246: Why was a Chezy coefficient of 65 not considered in the Terrebonne model? Are any time series available for comparison with the model?

As mentioned in a previous answer, this was a limitation of the model. Now, we have explored a wider range and found that result improve when 65 $m^{1/2}s^{-1}$ is used. We have also added a separate calibration with timeseries and validation using a different period. The validation also shows that the calibration with AirSWOT provided a model that is also temporally coherent because of the good agreement with CRMS timeseries. The result of the separate calibration and validation are reported in Figure S4.

Yes. Using 'southern' is more appropriate. We corrected.

Fig. 7: It would be helpful to add the date of the image in the figure (as in, e.g., Fig, 8) or in the caption. For the model calibration runs shown in this figure, what settling velocity was used for the results shown in Fig. 7A and what critical shear stress was used for the results shown in Fig. 7B?

We modified Figure 7. We have divided the figure into two new ones (Figure 7 and Figure 8 in the new manuscript). Date and fixed critical shear stress (for Figure 7) and settling velocity (for Figure 8) are now noted. Figures are also larger and clearer for the reader.

L271: Add a reference to Fig. 8 after noting the date/time of the imagery.

Reference added.

L274: Add reference to Fig. 8B after providing the RMSE of 35.88. Are there any in situ measurements of SSC available for checking model values?

Reference added. The new correct reference is Figure 10g. The evaluation of the AVIRIS-NG-based calibration was improved. Now there is a parallel calibration performed only with in-situ data. Then we also added a validation of both calibration using the Delta-X Fall campaign. We show that the calibration with AVIRIS-NG provided a better validation. We added two new figures in the supplementary material to show the new calibration (Figure S7) and the new validation (Figure S8). In the result we added a new paragraph that reads: "*Settling velocity calibrated using only in-situ measurements was found to be 0.325 mm/s, different from the value obtained from the remote sensing images (Figure S7). The optimal critical shear stress was identical (0.1 Pa). This combination provided the lowest RMSE of 25.26 mg/L. During the Fall campaign, when the two calibrations were compared, the calibration performed using AVIRIS-NG provided better agreement with in-situ measurements than the calibration with in-situ measurements only (Figure S8)*".

Also, in the discussion we added a passage discussion the results: "*A calibration using only measurements in the field presents limitations. The optimal set of parameters is $w_s$ = 0.325 mm/s, $\tau_{cr,e}$= 0.1 Pa (Figure S6h), which differs from the best set found using AVIRIS-NG $w_s$ = 0.25 mm/s, $\tau_{cr,e}$= 0.1 Pa). Interestingly, the validation shows that, despite the model tends to underestimate concentrations in both cases, the calibration with AVIRIS-NG provides the lowest RMSE (Figure S8). AVIRIS-NG provides the possibility to compare the results with spatially distributed data, allowing a more complete evaluation of model performance. Point measurements could also not cover the full range of concentration and lead to calibrated parameters not representative for the entire area. Especially in coastal areas, cohesive sediment properties are highly affected by flocculation. Fine sediment aggregate to form flocs, for which both settling velocity and bed shear strength are highly uncertain and difficult to predict*".

Section 4.1-4.3: Are there any reasons to think that the Chezy coefficient might not be uniform?

It is very likely that the Chezy coefficient is not uniform. The area is characterized by patched of *Spartina alterniflora,* thus is it likely to have variable friction. In our case, we are more interested in calibrating the friction. Zhang et al. (2022) suggest to first calibrating the topography using an initial uniform friction. Then calibrating the friction to account for vegetation. They suggest that because a direct calibration of the friction would lead to unrealistic distribution and values. Thus, for the purposes of our analysis, using a uniform friction is admissible. We specified this in the discussion of the UAVSAR coupling results.

L285: Does "lower half of the flight line" = "southern half of the flight line"? If so, it would be clearer to refer to southern half.

Yes. We clarified the expression using southern half.

L285-290: Could the comparison with Manning values be included in results instead of discussion, or is the Manning value so well known that the agreement is a useful point of discussion?

We moved the comparison with Manning in the results.

L306: "The validation of the water levels across the domain …further confirms the goodness of the calibrated friction coefficient" seems like an overstatement given that the previous paragraphs described portion of the model domain where the modeled and remotely sensed water levels do not agree – for reasons that may be unrelated to friction coefficient, but the disagreement makes it impossible to evaluate how well the friction coefficient works in those regions.

In the previous version of the manuscript we were not precise enough. In the new version we have added a new calibration using both AVIRIS-NG and timeseries from gauges, from both calibration and a validation in a different period (Delta-X 2021 Fall campaign) of the year, we show that AirSWOT leads to a better result. The tidal gauges are spread over the domain. The methods section was improved by adding the following paragraph: "*In the large-scale Terrebonne model, a second calibration of the Chezy coefficient and sediment properties was performed using only in-situ observations. Simulated water levels during the Spring campaign were compared with timeseries of 13 tidal gauges within the CRMS network (Figure S3). The RMSE and Nash-Sutclife Model Efficiency (ME) (Allen et al., 2007) were used to evaluate model performance for the different friction coefficients. The validation with timeseries allowed to evaluate the temporal coherence of the results that cannot be captured by remote sensing. The two calibrations were validated in the Fall campaign by comparing water levels. In the same model, a calibration of sediment parameters using in-situ measurements was carried out. Measured TSS concentrations from Fichot et al. (2022) were compared to simulation results (see Figure S5). Finally, the two calibrations were validated with in-situ TSS data collected during the Fall campaign (Fichot et al., 2022). RMSE was used to compare model results*". In the results section we point out that we obtained a better result with the AirSWOT-based calibration, while in the discussion we point out that there are still critical areas where we are not able to correctly simulate the water flow.

L316-327: The argument that friction plays w marginal role in affecting water levels on marsh platforms merits more discussion, since otherwise that seems like a reasonable alternative to arguing that marsh platform topography is poorly quantified. Is there a correlation between where the topography must be adjusted and vegetation characteristics on the marsh? Is it also possible that the model isn't resolving microtopography that affects water fluxes? How much was the topography altered to improve fluxes. Were the values realistic?

This is an excellent point that was not discussed in the previous version of the manuscript. The statement about the marginal role of the friction was not justified. This is a point that was tackled by Zhang et al. (2022) when the method was developed. The calibration of the topography inherently carries information on the friction. The authors ran a sensitivity analysis on the friction coefficient by testing different values (range 8-40 $m^{1/2}s^{-1}$), showing that a variation in friction has no effect on model performance. They point out that it is necessary to first adjust the topography. In the hypothesis of applying the calibration with UAVSAR only on friction, the final friction map would present unrealistic distribution and values because. Thus, they recommend to first calibrate the elevation and then the friction. We added this information in the discussion: "*As suggested by Zhang et al., 2022, friction plays a marginal role in affecting water levels on marsh platforms. They run a sensitivity analysis with a wide range of friction values and found little effect on model performance. The calibration of topography inherently contains information on friction, which can lead to its small effect on the computed flow field. Applying the same iterative method to friction only, without modifying marsh elevation, would lead to unrealistically large spatial variations of the friction coefficient. Therefore, it was decided to only change marsh elevation to match modelled water-level variations with those derived via UAVSAR*".
The modification of the topography produced realistic values. On average it was necessary to decrease the elevation by 1.3 cm with a standard deviation of 16.3 cm. Regarding the point on the microtopography effect on water fluxes, we added additional information, as it was worth of

discussion. The method depends on the ability of the model to solve the water fluxes. Therefore, if some areas are not well solved, it could be that the topography is changed even if it is not necessary. For instance, let's suppose there is a small channel that the 10 m mesh cannot represent. During falling tides, the areas near the channel drain faster, hence, they generate a larger water level change compared to internal areas. Since the model cannot solve the channel, the procedure will deepen the topography to allow water to flow and match the water level change, even if the topography is correct. We added this discussion by leveraging on some RTK measurements that we used to validate the calibrated topography: "*this iterative procedure might introduce errors in areas where the topography is correct. The methodology also depends on how well the model can solve the tidal channels. This effect can be noted for the three points before mentioned where the marsh was deepened. In this case, the points are located in proximity to a narrow channel with a 1.5-2 m cross section. The 10 m resolution of the model represents a limitation because features such as channels and levees that are smaller than 10 m cannot be captured by the mesh. In this example, the UAVSAR flight line captured the flow during falling tide, and in this phase, areas of the marsh close to the channels drained faster than internal ones. Since the model does not capture these channels, the method tries to compensate by lowering the marsh to increase water fluxes even if the elevation is correct*".

L333: change "worst" to "worse"

Corrected.

L337: Are any data available to evaluate this possibility?

Yes, there are other data available since there were repeat passes of the same flight line. Thus, with an extended analysis this aspect could be evaluated. Certainly, this is an interesting analysis that could be done in the future. For instance, it would be interesting to compare if the bathymetric correction changes (and how much) depending on the use of water level change during rising and falling tides. However, this goes beyond our scope, thus it is not included in the present analysis.

L349: Was TSS sampled just one time and at one location? If so, this doesn't seem like a robust enough test to declare the in situ sample to be more in error than the modeled value. What is the TSS value at that time and place in the AVIRIS-NG data? How much spatial and temporal variability do the model and remote sensing suggest?

There are many samples of TSS done at different times and well distributed in the domain. In the previous manuscript the evaluation of the calibration was not complete. Also, we were likely not clear. We are not suggesting that the modelled values are better than the in-situ sample. What we want to emphasize is that using spatially distributed data from remote sensing can provide greatly help improving the calibration because the model is compared with a large area instead of a few points that might not be representative for an entire region. In situ sample a still important because they are used to calibrate the models that derive these maps. The variability in the data is large especially when we compare the Spring with the Fall campaigns.

As indicate in previous answers now the calibration has been improved by including a second calibration with only in-situ data and a separate calibration in a different period with different measurements.

L356: Consider revising to "due to flocculation which increases settling velocities compare to …" In any case, the settling velocities used in the model for mud are effectively floc settling velocities.

We revised the sentence. It is correct, in the model these are flocs parameters. Indeed, our goal with the sentence is to remark that the process of flocculation introduces high uncertainty in the settling velocity and critical shear stress because we do not deal with single particles. The emphasis is on the flocs values. We recognized that in the previous version of the manuscript this was not clear. The sentence now reads: "*Especially in coastal areas, cohesive sediment properties are highly affected by flocculation. Fine sediment aggregate to form flocs, for which both settling velocity and bed shear strength are highly uncertain and difficult to predict*".

L360: Not clear what is meant by: "errors might be related to some the bathymetric modification"

This error is likely related to previous modifications of the text that were not corrected. We eliminated the 'the'. The sentence now reads: "*errors might be related to bathymetric modification of the mesh*".

L361: The channels were enlarged in the model? "might have generated"

Corrected.

L363: "inherent to the model. Although Delft-3D has 3-dimensional …"

Corrected.

L365-366: replace "tri-dimensional" with 3D and "bi-dimensional" with 2D.

We corrected all instances of 'tri-'and 'bi-dimensional' with 3D and 2D respectively.

L371: This paragraph might be better combined with the one before.

We agree with the suggestion as the two paragraphs are related. We merged them.

Fig. 9: It would be much better to use the same vertical scale for both profiles. Were the wave conditions much higher on Aug 19 (red curve) than on Aug 17? It seems important to recognize here that optical properties of sediment in suspension is strongly controlled by sediment size, and that the vertical profile of the finest, most optically active fractions might be more uniform that that of coarser bed fractions

We decided to eliminate Figure 9 from the manuscript. The purpose of the figure was to simply show that in some cases the sediment concentration in the water column in not homogeneous. This is certainly nothing new. In addition, all the discussion points hold without the need of the figure.

The last point raised by the Reviewer is certainly valuable. Indeed, in the case of a variable sediment concentration in the water column, the finer fraction is the one that is most likely resuspended to the surface. Hence, the finer fraction is what the sensor likely captures. This means that in some cases, AVIRIS-NG cannot capture the total sediment transport, since it does not see the coarser fractions that are located in the lower layers. We added a passage in the discussion that reads: "*Usually, the coarser fraction occupies the lower layers in the water column, while finer fractions are resuspended to the surface. Hence, since AVIRIS-NG can only characterize the water surface, TSS maps might not be representative of total transport, which represent a limitation for the calibration of sediment properties*".

Regarding the point on the optical properties as function of sediment size. In this specific situation, the connection between the surface sediments and optical properties is not straightforward. From the data collected it was not possible to retrieve a satisfactory relationship between the optical properties (e.g., absorption, backscattering) and grain size. One reason might be the limited range of diameters. The total range was 20-80 μm, with most of sediment falling within the 20-60 μm interval. Thus, given a wider range of diameters a relationship could be inferred, but for this specific case we cannot confirm it.

**References**

Feng, Z., Tan, G., Xia, J., Shu, C., Chen, P., & Yi, R. (2020). Two-dimensional numerical simulation of sediment transport using improved critical shear stress methods. *International Journal of Sediment Research*, *35*(1), 15-26.

Ganju, N. K. and Schoellhamer, D. H.: Decadal-timescale estuarine geomorphic change under future scenarios of climate and sediment supply, Estuaries and Coasts, 33, 15–29, https://doi.org/https://doi.org/10.1007/s12237-009-9244-y, 2010.

Liu, K., Chen, Q., Hu, K., Xu, K., and Twilley, R. R.: Modeling hurricane-induced wetland-bay and bay-shelf sediment fluxes, Coastal Engineering, 135, 77–90, https://doi.org/10.1016/j.coastaleng.2017.12.014, 2018.

Partheniades, E.: Erosion and deposition of cohesive soils, Journal of the Hydraulics Division, 91, 105–139, https://doi.org/10.1061/JYCEAJ.0001165, 1965.

Wiberg, P. L., Carr, J. A., Safak, I., and Anutaliya, A.: Quantifying the distribution and influence of non-uniform bed properties in shallow coastal bays, Limnology and Oceanography: Methods, 13, 746–762, https://doi.org/https://doi.org/10.1002/lom3.10063, 2015.

van der Wegen, M., A. Dastgheib, B. E. Jaffe, and J. A.Roelvink. 2011. Bed composition generation for mor-phodynamic modeling: Case study of San Pablo Bay in California, U.S.A. Ocean Dyn.61: 173–186.

Winterwerp, J. C., Van Kesteren, W. G. M., Van Prooijen, B., & Jacobs, W. (2012). A conceptual framework for shear flow–induced erosion of soft cohesive sediment beds. *Journal of Geophysical Research: Oceans*, *117*(C10).

Zhang, X., Jones, C. E., Oliver-Cabrera, T., Simard, M., and Fagherazzi, S.: Using rapid repeat SAR interferometry to improve hydrodynamic models of flood propagation in coastal wetlands, Advances in Water Resources, 159, 104088, https://doi.org/10.1016/j.advwatres.2021.104088, 2022.

---

## Author Response (AR2)

**Response to final comment**

We would like to thank again the Editor and the guest Associate Editor for handling our manuscript. We would like also to thank all reviewers for their time and constructive feedbacks.

We have corrected the error indicated by the Reviewer and guest Associate Editor. The sentence now reads: "*A Chezy coefficient of 65 m$^{1/2}$s$^{-1}$ returns a better match with the AirSWOT observations except for the third acquisition on 06 April 2010 UTC (Figure S2C), were slightly better results were found for a Chezy coefficient of 65 m$^{1/2}$s$^{-1}$*".

We ran another grammar check and found another error in the Acknowledgements. In the final sentence at line 472 we removed 'that', so the last part of the sentence reads: "*and all Delta-X team members that contributed to the conceptualization of the Delta-X mission and the data included in this study*".

**Colormaps and colors update**

As recommended, we have modified all figures to ensure readability for readers with color vision deficiencies. After the modifications, readability was tested using the open source Color Blindness Simulator by Matthew Wickline and the Human-Computer Interaction Resource Network at https://www.color-blindness.com/coblis-color-blindness-simulator/.

In Figures 2, 4, 5, 6, 7, 8, 9, and 10 the diverging 'RdBu' colormap was not modified because already colorblind safe. The major change was the switch from the 'turbo' to the 'viridis' colormap, but there were additional changes to other figures.

[Figure]

**MAIN TEXT**

| Figure | Modification |
|---|---|
| Figure 1 | We changed color of the domains with more distinguishable tones: **green**, **orange**, **purple**. |
| Figure 2 | Colormap in subplots (b) and (c) changed to the colorblind safe 'viridis'. In subplot (c), colorbar range was slightly modified from 0-150 to 0-130 mg/L for a better contrast. |
| Figure 3 | We changed the colors of the domain to be consistent with Figure 1 and changed the line type to a dotted line to better separate with flight-lines. Flight-lines |

| | extensions colors were changed ensure readability. **Pink** for UAVSAR, **yellow** for AirSWOT, and **green** for AVIRIS-NG. |
|---|---|
| Figure 4 | Colormap changed to 'viridis' |
| Figure 5 | Colormap changed to 'viridis' |
| Figure 6 | Subplots (f) and (g) colormap changed to 'viridis' |
| Figure 7 | Colormap changed to 'viridis' |
| Figure 8 | Colormap changed to 'viridis' |
| Figure 9 | Colormap changed to 'viridis' |
| Figure 10 | Colormap changed to 'viridis' |

**SUPPLEMENTARY INFORMATION**

Figures S1, S3, S5, S6, and S7 readability was verified and no change was necessary. Figures S4 and S8 were improved by changing **red** and **blue** with **green** and **orange**. In Figure S2, colormap was changed from 'turbo' to 'viridis'.